# Genetic aberrations in iPSCs are introduced by a transient G1/S cell cycle checkpoint deficiency

Ryoko Araki[1,5]*, Yuko Hoki[1,5], Tomo Suga[1,5], Chizuka Obara [1], Misato Sunayama[1], Kaori Imadome[1], Mayumi Fujita[1], Satoshi Kamimura[1], Miki Nakamura[1], Sayaka Wakayama[2], Andras Nagy [3,4], Teruhiko Wakayama [2] & Masumi Abe[1]*

A number of point mutations have been identified in reprogrammed pluripotent stem cells such as iPSCs and ntESCs. The molecular basis for these mutations has remained elusive however, which is a considerable impediment to their potential medical application. Here we report a specific stage at which iPSC generation is not reduced in response to ionizing radiation, i.e. radio-resistance. Quite intriguingly, a G1/S cell cycle checkpoint deficiency occurs in a transient fashion at the initial stage of the genome reprogramming process. These cancer-like phenomena, i.e. a cell cycle checkpoint deficiency resulting in the accumulation of point mutations, suggest a common developmental pathway between iPSC generation and tumorigenesis. This notion is supported by the identification of specific cancer mutational signatures in these cells. We describe efficient generation of human integration-free iPSCs using erythroblast cells, which have only a small number of point mutations and INDELs, none of which are in coding regions.

[1] Department of Basic Medical Sciences for Radiation Damages, National Institute of Radiological Sciences, National Institutes for Quantum and Radiological Science and Technology, Chiba 263-8555, Japan. [2] Advanced Biotechnology Center, University of Yamanashi, Kofu 400-8510, Japan. [3] Lunenfeld-Tanenbaum Research Institute, Mount Sinai Hospital, Toronto, ON M5G 1X5, Canada. [4] Australian Regenerative Medicine Institute, Monash University, Melbourne, Victoria 3800, Australia. [5] These authors contributed equally: Ryoko Araki, Yuko Hoki, Tomo Suga. *email: araki.ryoko@qst.go.jp; abe.masumi@qst.go.jp

A substantial number of point mutations have been reported in induced pluripotent stem cell (iPSC)[1–7] and nuclear-transfer embryonic stem cell (ntESC)[8–10] genomes and this could be causative for the tumorigenicity and immunogenicity of the derivatives of these pluripotent stem cells[11,12]. The molecular basis for these genetic aberrations is still largely unknown, and indeed it is still debated whether these mutations are pre-existing SNVs from the parent somatic cell population[3,13] or derived from iPSC generation events[4,7].

We revealed previously using whole genome sequencing (WGS) that the point mutations in mouse iPSCs show a transversion-predominant base substitution pattern[4]. Thereafter, a similar pattern was described in human iPSCs[5,6] and in another type of reprogrammed pluripotent stem cell, ntESCs[10]. Furthermore, variant allele frequency (VAF) analysis of these point mutations disclosed their heterogeneity within an iPSC clone, which was also observed in ntESC clones[4–6,10,14]. In other words, in addition to the SNVs which exist in all cells within a colony, a considerable number of SNVs was observed in only half of the clonal population. These observations clearly indicated that a substantial body of SNVs in iPSC and ntESC genomes cannot be explained by pre-existing SNVs and thus arose during iPSC generation.

In our present study, we attempt to elucidate the molecular basis for these iPSC generation-associated point mutations. Interestingly, a transient elevation of reactive oxygen species (ROS) at the early stage in iPSC generation was reported recently[15]. We speculated that this could be causative for transversion-predominant point mutations because an elevated ROS concentration efficiently produces 8-oxoguanine (8-oxoG) and 2-hydroxyadenine in DNA strands and/or in the nucleotide pool, resulting in transversion substitutions[16]. However, because 8-oxoG bases are usually corrected by the DNA repair system, the huge number of point mutations, several hundred to 1000/genome, detected in both human and mouse iPSCs derived from various types of somatic cells using various generation methods could not be readily explained in this way[4–6]. Namely, if repair systems and/or apoptosis systems operate normally, a cell bearing such a huge number of mutations would not be expected to survive. We here focus on the cell cycle checkpoints that function during genome reprogramming based on the hypothesis that these checkpoints are attenuated in a transient manner at the early stages of genome reprogramming. We first examine the sensitivity of iPSC generation to ionizing radiation (IR).

## Results

**Generation of human iPSCs with few genetic aberrations.** Here, we attempted to generate iPSCs using certain cells with low ROS concentrations. For this purpose, we focused on erythroblasts among the cell types that can be obtained less invasively. We prepared erythroblasts from human cord blood via expansion culture conditions which were optimized to enrich for this cell lineage[17] (Supplementary Fig. 1a). The subsequent analysis using CellROX Deep Red fluorescent probe revealed a lower ROS concentration in the enriched cells than in CD34-positive blood stem cells that are well known to be low-level ROS producers[18] (Supplementary Fig. 1b). Furthermore, a clear inhibition of pyruvate dehydrogenase (PDH), which catalyzes the conversion of pyruvate to acetyl-CoA and $CO_2$, and thereby links the glycolytic pathway to the tricarboxylic (TCA) cycle, was clearly evident in CB-erythroblasts. The phosphorylation of the three Ser residues in PDH, S232, S293, and S300, indicating the inactivation of this enzyme, was detected in these cells and PDK3 seemed to be the responsible kinase (Supplementary Fig. 1c and d). These results indicate that the energy metabolism of CB-erythroblasts is

glycolytic-dominated and is suppressed in the mitochondria, which are the major producers of ROS (Supplementary Fig. 1e). Furthermore, our analysis revealed a remarkably high expression of antioxidant genes in erythroblasts (Supplementary Fig. 1f).

We employed five episomal vector-cloned genes, Oct3/4, Sox2, Klf4, c-Myc, and Lin28 for iPSC generation (Supplementary Fig. 2a)[17], and thereby successfully established 14 independent iPSC clones (CB-erythroblast–iPSC clones) from four individuals in total (5, 3, 3, and 3 independent iPSC lines from donors 1 to 4, respectively; Fig. 1a, b). We next performed WGS of these 14 clones and compared them with the data reported for human iPSCs thus far. Of note, we adjusted for possible bias from the informatics process by downloading prior iPSC WGS data from a public database and reanalyzing this information using our bioinformatics system (Supplementary Fig. 2b and c). As anticipated, a dramatic decrease in the number of point mutations was evident for all the 14 clones (Fig. 1b and Supplementary Data 1). Importantly, half (7/14) of the CB-erythroblast–iPSC clones in our panel showed no amino acid substitutions in their gene coding regions, 5/14 showed no point mutations at all and 2/14 showed synonymous substitutions (Fig. 1c and Supplementary Table 1). In addition to point mutations, we have established a detection system for genomic INDELs (insertions/deletions) and observed a marked decrease in their frequencies in all of the aforementioned 14 iPSC clones (Fig. 1d, Supplementary Data 1 and 2). We conducted a confirmation test for 23 INDEL candidates detected in four iPSC lines (#1–1, 1–5, 1–8 and 1–11) from donor 1. We successfully confirmed 19 out of these 20 candidates using Sanger sequencing (Supplementary Fig. 3a and b). The PCR reaction failed in three of the INDEL candidates. Furthermore, we also analyzed copy number variations (CNVs) in six iPSC lines using a high-resolution SNP array using two of the parental cell populations as controls. Only one CNV was detected in the CB-epi-1-8 iPSC line. (Supplementary Fig. 4) which notably contained a non-coding transcript, LINC00333, of unknown function [https://www.genecards.org/cgi-bin/carddisp.pl?gene=LINC00333]. Thus, CB-erythroblast–iPSCs largely solve the genetic aberration problems with these stem cells, which is one of the principal concerns with regard to their possible future medical applications. The integration-free status of these clones was confirmed by both PCR (Fig. 2a) and WGS (Supplementary Fig. 5). The pluripotency of these cells was also definitively verified by assaying for stem cell marker products (Fig. 2b) and developmental abilities using embryoid body (Fig. 2c). In the following sections, we describe the molecular basis for the onset of point mutations in iPSCs that prompted us to adopt our current approach to solving this issue, i.e. our selection of erythroblasts as the parent somatic cells for generating genetic burden-free iPSCs.

**The early stages of iPSC generation are IR resistant.** To elucidate the molecular basis of the point mutations identified in iPSC genomes, we hypothesized that the normal cell cycle checkpoint function in cells is transiently attenuated during iPSC generation. To test this possibility, we first examined the sensitivity of iPSC generation to X-ray irradiation, particularly during the early stages. We counted iPSC colonies after a single dose of 3 Gy irradiation on days 1–6 post-infection when using an iPSC generation assay with Nanog-GFP MEFs infected using Yamanaka 4 factors, Oct3/4, Sox2, Klf4, and c-Myc expressed in retroviral vectors[19,20]. Although a reduction in colony frequency was observed at day 1 and days 4–6 following irradiation as anticipated, higher frequencies of colony formation compared with the controls were observed on days 2 and 3 (Fig. 3a, Supplementary Figs. 6, 7a, and b). Because only 10–40% and 2–60% of MEFs and

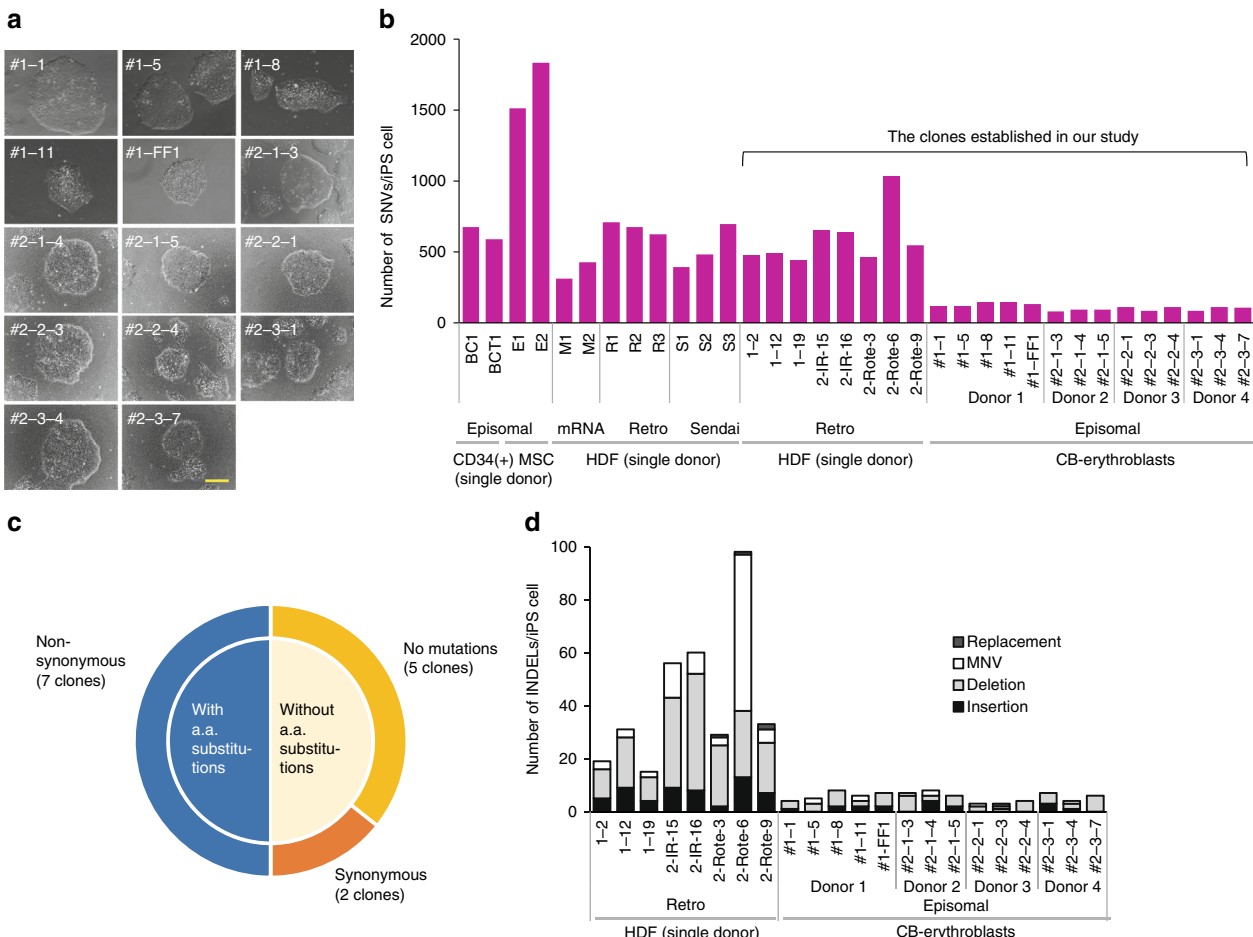

**Fig. 1 Generation of human iPSCs harboring fewer point mutations. a** Morphology of the established human iPSCs generated from the erythroblast-rich fraction expanded from cord blood. The asterisk denotes 'CB-epi-' (e.g. #1–1 indicates clone CB-epi-1-1). Scale bar, 300 μm. **b** The number of SNVs observed in various human iPSC clones. Twenty-two lines were established in this study including 14 genome integration-free human iPSC lines derived from four different donors and eight lines generated by retroviral gene transduction from human dermal fibroblasts (HDF). In addition to our human iPSC clones, we re-analyzed the WGS data for 12 previously reported human iPSC clones: BC1, BCT1, E1, and E2[2,75] and M1, M2, R1, R2, R3, S1, S2, and S3[5]. All samples in the present study were subjected to identical bioinformatics analysis to minimize possible differences in the SNV identification process from alternate methodologies (Supplementary Fig. 2b). Note that 5 out of the 8 HDF-derived iPSC lines (2-IR-15, 2-IR-16, 2-Rote-3, 2-Rote-6, and 2-Rote-9) were additionally treated with external stimuli during their generation. IR, ionzing radiation; Rote, rotenone (Supplementary Data 1 and Supplementary Fig. 22). **c** Seven out of 14 CB-erythroblast–iPSC clones in our panel showed no amino acid (a.a.) substitutions in their gene-coding regions, 5/14 showed no point mutations at all and 2/14 showed synonymous substitutions (see Supplementary Table 1 for more details). **d** A marked decrease was observed in the INDEL frequencies in all of the aforementioned 14 iPSC clones from erythroblasts. Replacement: e.g. A > CT, multiple nucleotide variants (MNV): e.g. GA > TT. See Supplementary Data 2 for more details.

ES cells can survive after 3 Gy irradiation, respectively[21–27], the results on days 2–3 were really surprising.

To validate these results, we investigated cell cycle check point activities at days 2–3 using the checkpoint inhibitor wortmannin, which blocks the PI(3)K (phosphatidylinositol-3-OH kinase)-like kinases such as ataxia telangiectasia mutated (ATM) and ataxia telangiectasia and Rad3 related (ATR), that play central roles in cell cycle checkpoints, in a dose-dependent manner[28,29]. This inhibitor is thus widely used in cells as an effective sensitizer to radiation[30]. We assessed the frequency of iPSC generation irradiated with 3 Gy followed by growth in culture medium supplemented with Wortmannin at 10 μM for 24 h, which is within the inhibition range for ATM but not for ATR (Supplementary Fig. 8 left)[31]. A considerable effect was evident on days 4 and 6 as anticipated, but little or no sensitizing effect was detected on days 2 or 3, indicating that the cell cycle checkpoint was fully functioning on day 6, but not on days 2 or 3. In addition, to exclude the possibility that medium replacements

in our culture affect the sensitivity of wortmannin, we conducted the same experiment using modified culture conditions [2] (Supplementary Fig. 6), in which there was no medium replacement from the onset of reprogramming until day 7. This approach produced similar results, i.e. a strong sensitizing effect on day 6 but not on days 2 or 3 (Supplementary Fig. 8 right). This was done to clarify whether the changes in sensitivity was dependent on culture conditions.

To confirm our observations regarding cell cycle checkpoint activities, we performed cell cycle analysis via flow cytometry at 24 h-post 3 Gy irradiation at day 3 and day 6 (Fig. 3b). We found from this analysis that at day 3 the cell cycle showed an absence of arrest at 24 h post 3 Gy irradiation that was similar to the control cells without IR (0 Gy). In contrast, at day 6, cell cycle arrest was evident post 3 Gy irradiation. It is also noteworthy that a marginal degree of arrest occurred in the control cells without IR also due to the confluent state of these cultured cells. For this experiment, we used a Dox-inducible iPSC generation system, for

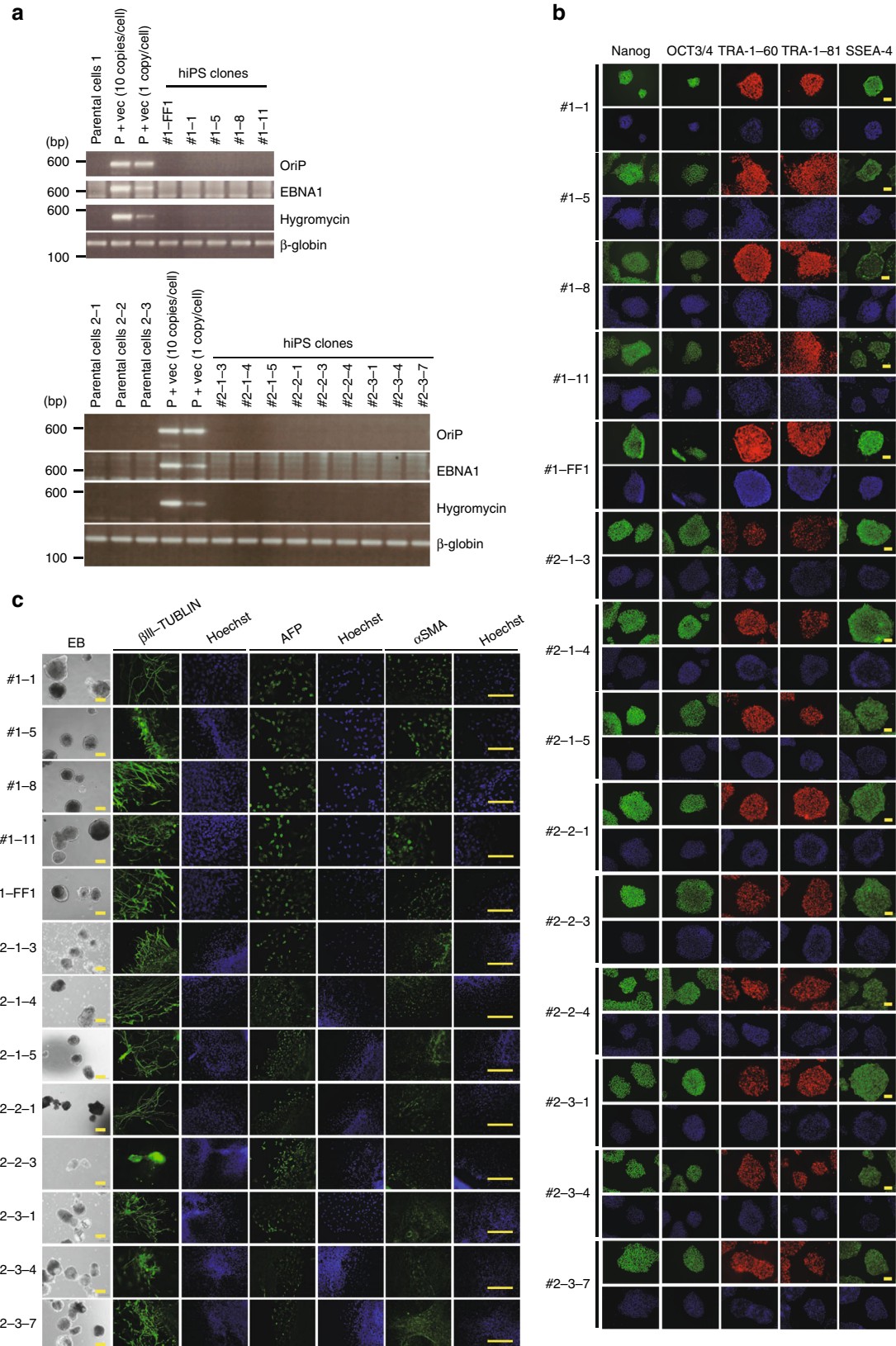

which the experimental outline and additional results are shown in Supplementary Fig. 9. These results indicated that the G1/S checkpoints were seriously deficient in these cells at day 3. Furthermore, we performed an Annexin V assay to evaluate apoptosis activities and observed a clear difference between day 3 and day 6. The results in this case showed a high induction of apoptosis in day 6- but low induction in day 3-irradiated cells (Supplementary Fig. 10). Hence, our analyses of the radio-sensitivity of iPSC generation and subsequent evaluations suggested a transient reduction in cell cycle checkpoint activities in the very early stages of iPSC generation, i.e. on days 2–3 post-infection.

**Fig. 2 Pluripotency and differentiation ability of iPSCs derived from CB-erythroblasts. a** Confirmation of an integration-free status in the established iPSCs. The positions of the PCR amplicons for OriP, EBNA1, and hygromycin in the plasmid are shown in Supplementary Fig. 2a. Template mixtures, 10 copies of pEB-C5 vector/cell, and one copy of pEB-C5 vector/cell, were used as positive controls. Genomic DNA corresponding to ~10,000 cells was used in each sample, β-globin was used as the control for the presence of genomic DNA. **b** Immunocytochemical staining of stem cell markers is shown with Hoechst33342 counterstaining (bottom of each panel). Scale bar, 100 μm. **c** Immunocytochemical staining of tissue-specific markers was conducted to assess the abilities of the iPSCs to differentiate into three germ layers. βIII-tubulin, alpha-fetoprotein (AFP), and α-smooth muscle actin (αSMA) were used as markers of ectoderm, endoderm, and mesoderm, respectively. Hoechst33342 counterstaining is also shown. Scale bar, 200 μm (EB) and 100 μm (immunofluorescence). EB, embryoid body.

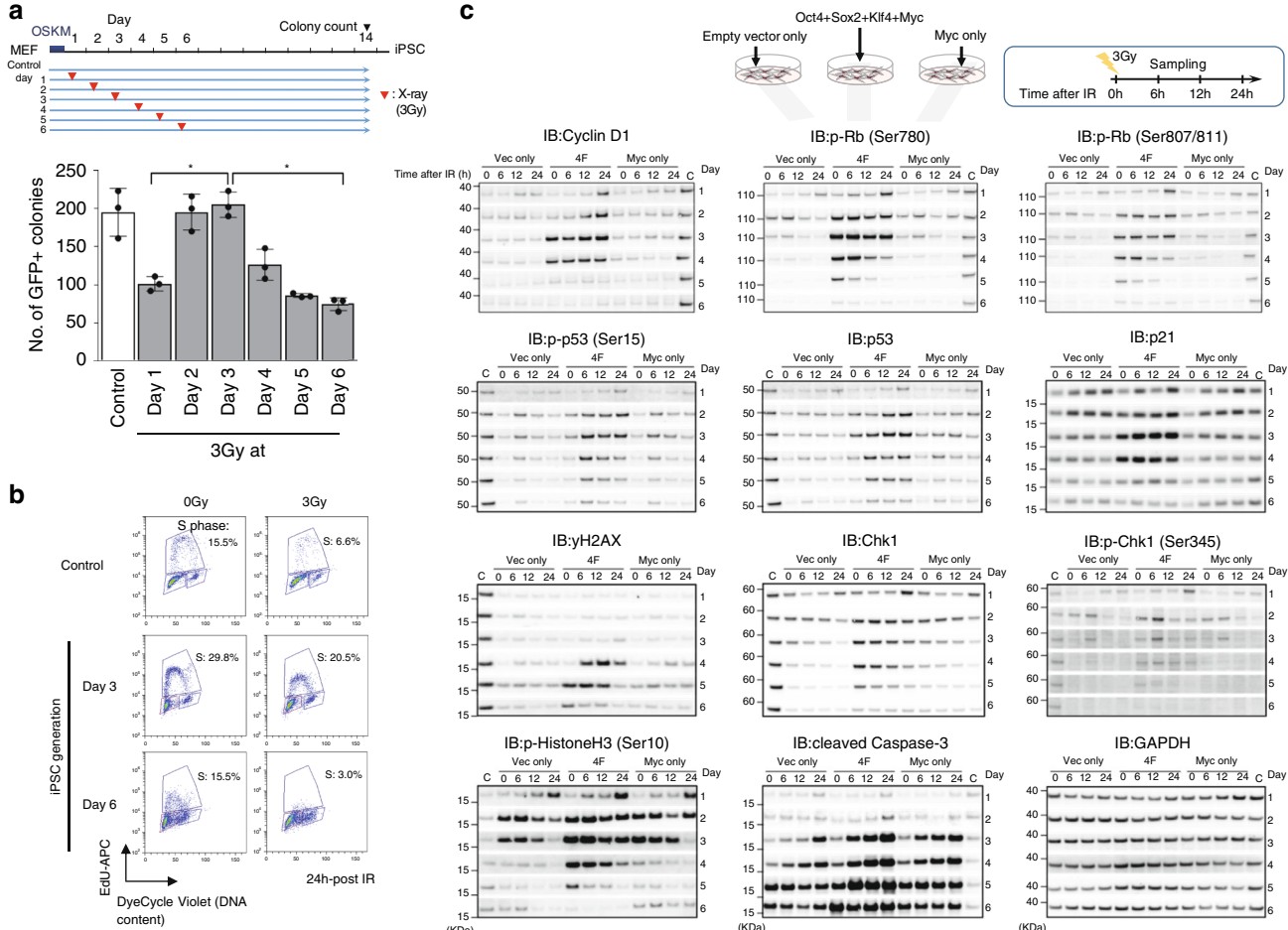

**Fig. 3 Radiosensitivity of iPSC generation. a** Nanog-GFP MEFs infected with Yamanaka 4 factors (OSKM) were exposed to 3 Gy X-ray irradiation on each day post-infection up to day 6 (culture condition [1]. See Supplementary Fig. 6). GFP+ (Nanog+) colonies were counted on day 14. Error bars denote the standard deviation (SD) of the mean ($n = 3$ independent culture dishes). *$P < 0.05$. Results are representative of three independent experiments with three replicates per group. Control, without ionizing radiation (IR). For statistical analysis, two-sided Mann–Whitney $U$ test was performed. **b** Cell cycle analysis using EdU and propidium iodide (PI) in a Dox-inducible system. The cells used were MEFs derived from a chimeric mouse generated with a GFP-positive iPSC line containing the Dox-inducible transgene constructs encoding the four reprogramming factors (see Supplementary Fig. 9a). Control cells were doxycycline-untreated GFP-negative normal primary fibroblasts prepared from the chimeric embryos. To control for possible effects of culture medium replacement, this analysis was conducted with no replacement of the growth medium throughout the iPSC generation process from Dox induction (culture condition [4]). Notably, similar results were also obtained using original culture conditions [3]. **c** Western blot analysis of cell cycle-related proteins on days 1–6 using whole cell lysates prepared at 6, 12, and 24 h after 3 Gy irradiation (MEF, 4F retroviral system, culture condition [1]). Two different controls were used as follows: empty vector infection alone (Vec only) and c-Myc infection alone (Myc only). Cell extracts were prepared each day at 6, 12, and 24 h after 3 Gy irradiation. The signals normalized to the GAPDH signal are also shown (Supplementary Fig. 12). In addition, because the sample set for each day was analyzed with different western blots, an internal control was loaded to normalize the exposure time. Source data are provided as a Source Data file.

**Molecular basis of the transient G1/S checkpoint deficiency.** To next explore the molecular basis of our findings, we investigated the key factors involved in cell cycle checkpoint functions including p53, cyclin D1, Rb, and Chk1, and found that they are expressed at remarkably different levels between day 3 and day 6 (Supplementary Fig. 11a). This indicated an elevation in checkpoint-related activities at around day 3. Thereafter, we conducted further analysis using the samples consecutively prepared on each day from 1 to 6 to obtain more conclusive data (Fig. 3c). In addition, we employed two controls, vector

only and c-Myc only infection, to uncover reprogramming-specific but not c-Myc-specific phenomena. Surprisingly, our results indicated a clear and profound boost in the cyclin D1 level, by over 100-fold, on days 3 that was suddenly down-regulated at the end of day 4 (Fig. 3c and Supplementary Fig. 12 left upper). Intriguingly, this expression profile was unaffected by 3 G irradiation. This indicated a 4F-specific phenomenon that was not due to c-Myc transduction as no such induction was observed in either the vector or c-Myc only infection controls. Importantly, the stage-specific boost of cyclin D1 was strongly supported by the phosphorylation of Rb that was demonstrated using two different anti-Rb phosphorylation antibodies (anti-Ser780 and anti-Ser807/811). More importantly, the phosphorylation of Rb was not downregulated by IR at days 2–3. In contrast, a reduction of Rb phosphorylation in response to IR was observed at days 4–5. Hence, our western blot analysis revealed the potent but transient activation of the cyclin D1-Rb phosphorylation pathway during iPSC generation leading to a cell cycle checkpoint deficiency, especially at G1/S, at around day 3 in a transient manner. Our findings were also consistent with the strong and IR-insensitive histone H3 phosphorylation signals that we observed, which are a good indicator of M phase.

We also investigated genome damage-sensing factors, such as p53 and Chk1, including their phosphorylation status, and observed similar profiles for IR although the induction of these proteins was retained until day 5, suggesting a feedback loop from the enhancement of cyclin D1-Rb phosphorylation signals to p53-related pathways (Fig. 3c and Supplementary Fig. 13)[32]. A rapid upregulation of p53-related activities following cyclin D1-Rb phosphorylation induction may cause a swift quenching of cyclin D1 functions, thereby restoring the activated state of the cell cycle checkpoint through the induction of cyclin D1 by OSKM. Indeed, we found in our analysis that the increase of p53-related functions resulted in a rapid enhancement of γ-H2AX activity post-3Gy irradiation on day 4 (Fig. 3c); the γ-H2AX response to IR was not seen until the 3rd day and was first observed on the 4th day. In addition, western blot and real-time PCR analyses of p21 strongly suggested an unresponsiveness to IR at day 3 (Fig. 3c and Supplementary Fig. 14).

We obtained additional supporting results when we conducted experiments using the nucleotide analog, 5-ethynyl-2′-deoxyuridine (Edu) (Supplementary Fig. 15). Edu is widely used for in vivo labeling as well as BrdU, but its considerable toxicity has also been reported[33]. Indeed, when Edu was added for 12 h to our cultures at day 3, almost no colonies appeared in any instance, with or without IR. In contrast, 3 Gy irradiation prior to Edu incubation at day 6 rescued iPSC generation, although a substantial number of colonies also appeared without IR, possibly due to the restoration of cell cycle checkpoint activity followed by DNA repair systems. This indicated that the DNA repair machinery is active at day 6 and that a cell cycle arrest in response to IR has occurred at this timepoint resulting in an Edu incorporation block. These data strongly suggested a deficiency in both cell cycle arrest and subsequent DNA repair functions at day 3, thereby enabling an efficient incorporation of modified/damaged nucleotides at this timepoint.

Taken together, we conclude from these data that a reduction in the G1/S cell cycle checkpoint, resulting in a DNA repair and apoptosis deficiency, occurs in a transient manner on days 2–3 during iPSC generation through the competitive inhibition of the p53-Rb pathway via cyclin D1 overexpression (OE), induced by OSKM-reprogramming factors. We therefore propose a transient radio-resistance and G1/S checkpoint deficiency model based on the mechanisms revealed by our present findings (Supplementary Fig. 13).

**Radio-resistance phenomenon in other iPSC generation systems**. We revealed in our current analyses using a retroviral gene transduction system that there is an attenuation of G1/S cell cycle checkpoint activity in the early stages in iPSC generation. To confirm that this finding was not due to the use of the retroviral vector, we also examined iPSC generation in response to IR using a Dox-inducible mouse iPSC system[34,35]. We obtained similar results to those seen with the retroviral system; iPSC generation exhibited an IR-resistant phenotype in its early stages at days 2–4 after Dox induction (Fig. 4a, Supplementary Figs. 16 and 17a). Importantly, similar results were also found during human iPSC generation; two types of human iPSC generation systems were examined (Fig. 4b and Supplementary Fig. 17b for the Sendai viral system, and Fig. 4c for the retroviral system). Notably in the experiment using retroviral system, we used modified culture systems without medium replacement, at least from day 0 to day 7, to exclude the possible effects of this replacement on radio-sensitivity during iPSC generation (Supplementary Fig. 6). Cell cycle analysis of human iPSC generation supported these findings (Fig. 4d). Lastly, we investigated 3F (without c-Myc)–iPSC generation, because c-Myc over expression could increase the tolerance of cells to IR, and also observed radiation-resistant phenotype (Fig. 4e). Although we observed a rather delayed and broad peak compared to 4F-iPSC generation, this was consistent with previous observations on 3F-iPSC generation[36,37].

In addition, we identified using western blot analysis that cyclin D1-Rb phosphorylation activity is enhanced in these iPSC generation systems (Supplementary Fig. 11b and c). Thus, our results were not dependent on the procedure nor on the cell type used for genome reprogramming. However, it is noteworthy that no difference in p53-related activities, other than Chk1 products, was observed between day 3 and day 6 during human iPSC generation (Supplementary Fig. 11c).

**Mutation generation is limited to a few cell divisions**. Our present analyses were based on the hypothesis that a transient attenuation of DNA repair functions occurs during iPSC generation and is the mechanism underlying the occurrence of point mutations in the iPSC genome. Our subsequent findings using an IR-sensitivity assay and subsequent analyses indicated a defect in the normal cell cycle checkpoints at the very early stages of iPSC generation, i.e. days 2–3 post-infection by Yamanaka factors. Of note also, the abnormality revealed in the early stage of iPSC generation, i.e. the G1/S checkpoint deficiency, closely resembles that observed in various cancers[38]. We investigated therefore whether the point mutations observed in iPSC genomes arose within a limited period during days 2 to day 3 post-infection.

While it has been considered that point mutations are basically distributed over all cells within an iPSC colony, recent VAF analysis revealed that a considerable number of SNVs are present in only a portion of the cell population in a colony[4,6,10]. Importantly, the VAF information for each SNV also provides a timeline for their emergence and we succeeded previously in measuring the occurrence of point mutations at the initial stages of iPSC generation (Supplementary Fig. 18a)[4]. Meanwhile, it must be noted that we could not determine whether such mutations occur continuously or transiently, namely throughout the whole process of iPSC generation or during a specific stage only.

Current WGS analysis of iPSC genomes basically detects only highly frequent SNVs, but cannot detect the low or even moderate frequency SNVs that are essential for determining the point mutation accumulation throughout the iPSC generation process. In essence, a standard WGS method basically identifies 50%VAF-SNVs, i.e. wild/mutant in all cells within an iPSC

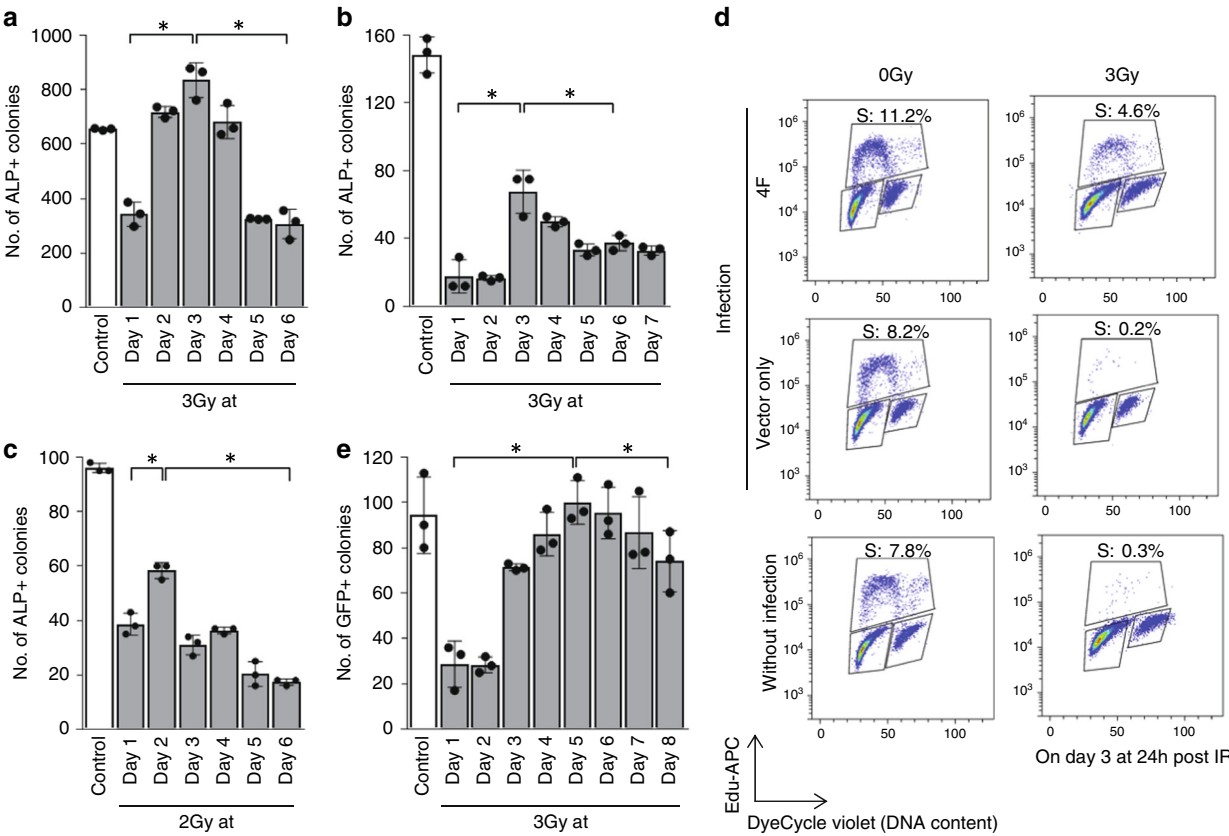

**Fig. 4 Verification of stage-specific radio-resistance in iPSCs generated using different systems. a** Dox-induced MEFs were exposed to a 3 Gy dose of X-ray irradiation on days 1–6 and ALP+ colonies were counted on day 12 (Supplementary Fig. 6 culture condition [4]) (means ± SD, $n = 3$ independent dishes). Results are representative of two independent experiments with three replicates per group. Similar results were also obtained using original culture conditions [3]. **b** Human BJ fibroblasts were infected with Yamanaka 4 factors (OSKM) using Sendai virus vectors and then exposed to 3 Gy X-ray irradiation (IR) on days 1–7. ALP+ colonies were counted on day 12 (culture condition [5]) (means ± SD, $n = 3$ independent dishes). **c** BJ fibroblasts infected with five factors (OSKNL) using retroviral vectors were exposed to 2 Gy of X-ray IR on days 1–6 and ALP+ colonies were then counted on day 25 (culture condition [7]) (means ± SD, $n = 3$ independent dishes). Results are representative of two independent experiments with three replicates per group. **d** Cell cycle analysis using Edu was performed at 24 h post 3 Gy irradiation, and in cells without IR (0 Gy), on day 3 and day 6 of iPSC generation (BJ fibroblasts, retroviral system) (culture condition [7]). Data are representative of three replicates. **e** Nanog-GFP MEFs infected with three factors (OSK) were exposed to 3 Gy X-ray irradiation on days 1–8, and GFP+ (Nanog+) colonies were then counted on day 15 (culture condition [9]) (means ± SD, $n = 3$ independent dishes). Results are representative of two independent experiments with three replicates per group. For statistical analysis, two-sided Mann–Whitney $U$ test was performed for all statistical analyses. *$P < 0.05$. Source data are provided as a Source Data file.

colony. Here, we conducted deep WGS of eight iPSC lines to detect lower frequency VAF-SNVs in addition to 50%VAF-SNVs (Supplementary Fig. 18b). The total sequences obtained were ~240 Gb (R-4F-535, 536, 539, 541, 594, and 596), which correspond to a two-fold higher coverage of the genome than previous WGS analyses[5,6]. De novo SNVs were exclusively identified in accordance with a previously reported procedure for which reliability has been confirmed[4]. As a result, we revealed a significant number of SNVs in the genomes of these lines, 744–1665, and intriguingly, found another peak or shoulder in addition to that for the 50%VAF-SNVs which form the main peak in such clones. Five clones, R-4F-13, 17, 535, 536, and 539, had additional peaks at around 20–30%VAF, and clone 541 showed a shoulder shape, suggesting the presence of large numbers of SNVs of <50% VAF (Supplementary Fig. 18b).

To overcome the technical hurdle that prevents the detection of lower frequency SNVs, we adopted a new approach. WGS was performed using lines derived from single cells prepared from identical iPSCs (hereafter referred to as sublines) to identify SNVs comprehensively (Fig. 5a and Supplementary Fig. 18b). Because this method enables us to identify SNVs regardless of their VAF in an iPSC colony, SNVs bearing a VAF from 50% to an

extremely low frequency can be detected at around a 50% allele frequency in subline genomes. Indeed, with this method we successfully identified 2–4 fold more mutations compared to the usual WGS analysis of an iPSC genome. As anticipated, amplicon sequencing clearly revealed that all SNVs with a VAF = ~50% were detected in the sublines, but never in the parent somatic cell genome indicating that these SNVs are not pre-existing SNVs (Supplementary Fig. 18c and d). In our current analysis, we used sublines for which a developmental tree had either been previously described or newly determined (Supplementary Fig. 18e and Supplementary Data 3)[4].

Using the aforementioned approach, we analyzed three different types of reprogrammed pluripotent stem cell lines: R-4F-17 and 28 (retroviral iPSCs), 2A-4F-136 (integration-free iPSC), and B6M3-ntES-1 (ntESC). The results of WGS were then displayed using a Venn diagram and developmental trees (Fig. 5b). A substantial number of SNVs that were present in only a portion of the cell population of a colony were detected in every reprogrammed stem cell line. Furthermore, we revisited the WGS data of our iPSCs and ntESCs, from which single cell-derived sublines were established, to investigate how many low and extremely low VAF-SNVs could be identified by direct WGS

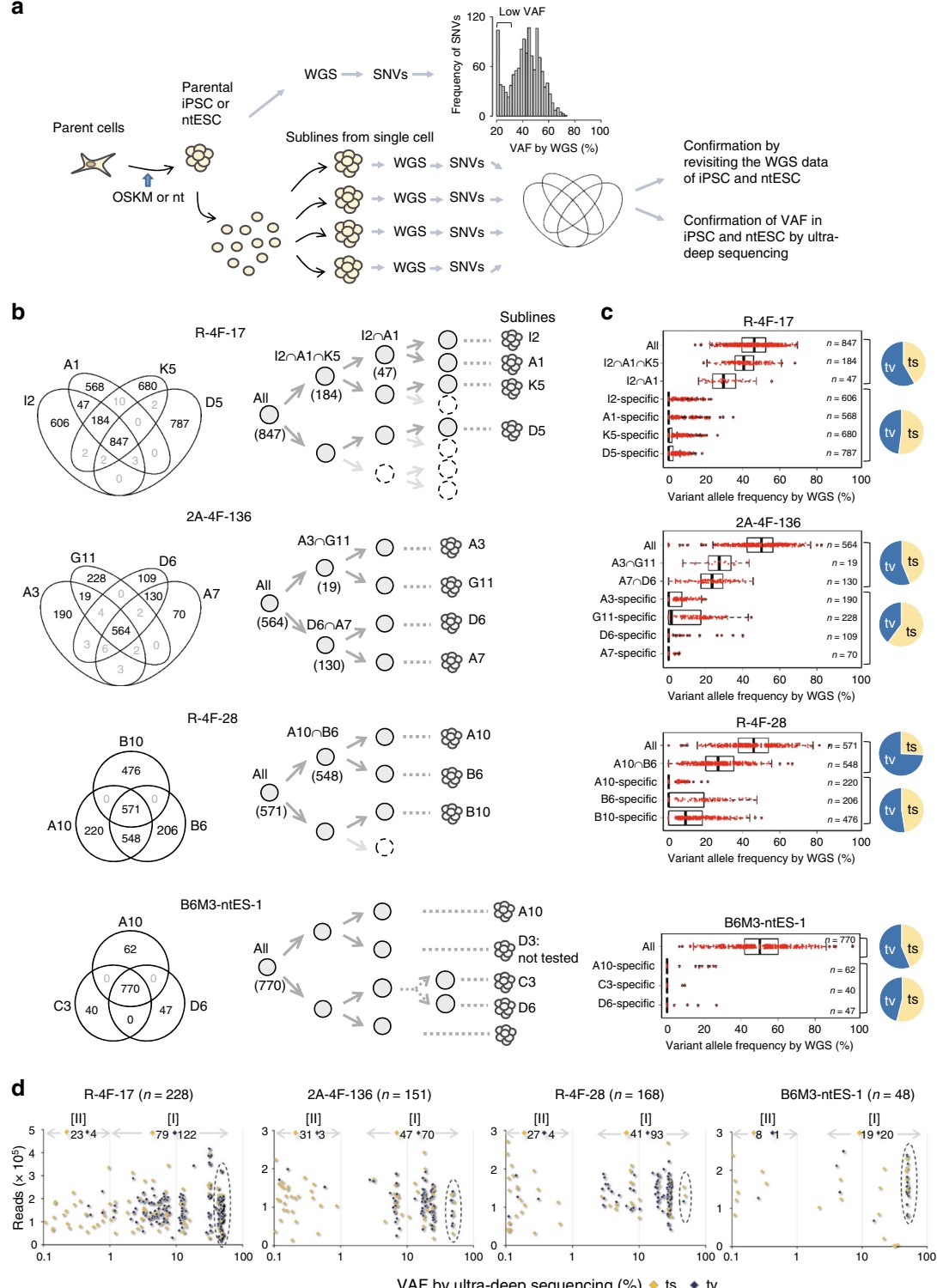

of reprogrammed stem cell genomes. These data clearly revealed that SNVs identified by conventional WGS analysis of iPSCs and ntESCs consist of various VAF-SNVs (Fig. 5c). Notably, there was a difference found between iPSCs and ntESCs in terms of the mode of SNV development: iPSCs but not ntESCs show a considerable number of low-frequency SNVs.

Finally, we conducted ultra-deep-sequencing analysis of the amplicons to more precisely determine the actual allele frequency of the <50%VAF-SNVs identified in sublines in the parent-reprogrammed pluripotent stem cells, and assess the precise

timeline of their emergence (Supplementary Fig. 18f). Because amplicon sequencing allows us to obtain ~10^5 reads for each SNV, ~10,000 times more than the reads obtained by standard WGS, a precise and highly sensitive VAF measurement became possible. The reliability of our analysis was verified in a control experiment, the predicted and actual results of which are shown in Supplementary Fig. 19a. We evaluated VAFs identified by WGS analysis of various sublines (Fig. 5d). We randomly chose a number of SNVs that are shared among several, although not all, sublines: 2A-4F-136 (number of SNVs examined, 214), R-4F-28

**Fig. 5 Identification of low-frequency SNVs using WGS analysis of sublines established from single cells of parental iPSCs and ntESCs populations.** **a** Conceptual flow: comprehensive identification of SNVs by WGS of individual cell-derived sublines and investigation of their allele frequencies (VAF) in the parental iPSC clones by ultra-deep (amplicon) sequencing, and by revisiting the WGS data from the parent stem cells. **b** Shared SNVs and subline-specific SNVs are shown in the Venn diagram (left). The timeline of the appearance of SNVs is shown in the developmental tree (right. See also Supplementary Fig. 18e and Supplementary Data 3 for further details). **c** We revisited the conventional WGS data obtained from the parent reprogrammed stem cells and evaluated whether SNVs identified via subline WGS were detectable also by conventional WGS. The results are shown in the Box-plot. Boxes indicate median values with the 25th and 75th percentiles and whiskers show the 1.5 interquartile range. *n* = number of SNVs that were identified by subline analysis as mentioned above. Although most shared SNVs had been detected by WGS analysis of the parent stem cells, not many subline-specific SNVs were identified by this conventional method. ts transition, tv transversion. **d** Ultra-deep VAF analysis (amplicon sequencing) of parental reprogrammed stem cells for each SNV was performed to investigate the actual VAF in the corresponding parent stem cells. Three types of reprogrammed stem cells including two retroviral iPSC lines (R-4F-17 and R-4F-28), one integration-free iPSC line (2A-4F-136) and one ntES line (B6M3-ntES-1) were examined. The *X*- and *Y*-axes indicate the VAF and sequence redundancy, respectively. The broken circles denote the 50% SNVs. *n* = the number of SNVs for which data are shown. Note that SNVs of <0.1%VAF are not shown because they could not be distinguished from the background signal (Supplementary Fig. 19a sample 6). The base substitution of each SNV, ts (orange diamond) or tv (black diamond), is shown and their total numbers are also indicated for each category.

(number of SNVs examined, 216), R-4F-17 (number of SNVs examined, 259) and B6M3-ntES-1 (number of SNVs examined, 73).

Two groups emerged from this: group I containing ~5–50% VAF SNVs and group II comprising SNVs of a less than ~1% VAF. Details of the results for each subline are shown in Supplementary Fig. 19b. Importantly, a remarkable difference in the base substitution pattern between these two groups was also clearly evident. Group I exhibited a transversion-predominance, whereas almost all of the SNVs categorized in group II were transition base substitutions (Fig. 5d). These data were consistent with those shown in Fig. 5c. Moreover, and intriguingly, the mode observed for group I SNVs in terms of frequency and concentration was different from that found for group II, indicating a transient and not continuous occurrence of point mutations within a limited number of cell divisions (Fig. 5d and Supplementary Fig. 18f).

Group I SNVs thus represent iPSC generation-associated point mutations, which arise in a concentrated fashion within only a limited number of cell divisions at the very early stages of iPSC generation. In contrast, group II SNVs may represent spontaneous point mutations. Similar profiles were also observed in all reprogrammed pluripotent stem cells that were tested, including retroviral iPSCs, integration-free iPSCs generated using the 2A-plasmid vector and ntESCs, suggesting a generality of our present findings in terms of genome reprogramming processes (Fig. 5c, d).

Although almost all of the 50%VAF-SNVs identified by our system could not be detected in the corresponding parent somatic MEF genomes even by ultra-deep sequencing[4] (Supplementary Fig. 18d), we still cannot completely exclude the possibility that these SNVs had existed in their parent somatic cells at frequencies below the detection limit of the methodologies we used. Therefore, we employed a new approach using single cells from a reprogrammed cell clone to identify bona-fide de novo SNVs, because these need to be distinguished from pre-existing SNVs when elucidating genome reprogramming-associated point mutations (Fig. 5a). WGS analysis of single cell-derived subclones revealed a substantial number of SNVs with a <50%VAF that definitely emerged after the reprogramming process (Fig. 5b). We focused on these bona-fide de novo SNVs in our current analysis and thereby revealed the timeline of the SNV emergence in reprogrammed stem cells.

**Mutational signatures associated with genome reprogramming.** Because a cell cycle checkpoint deficiency is also crucial for tumorigenesis and is a typical characteristic of cancer cells, although not a transient property, we performed cancer mutational signature analysis of our stem cells. This assay was

developed to analyze the huge body of cancer sequence data that has been accumulating rapidly due to next-generation DNA sequencing, and has successfully identified a number of cancer type-specific mutational signatures to date [https://cancer.sanger.ac.uk/cosmic/signatures_v2]. In our current study, we wished to explore whether genome-reprogramming processes share any pathways in common with tumorigenesis.

We identified a known cancer signature, Signature 17 (involving a high frequency of 'T' to 'G' transversions at CTT, GTT, TTT and ATT sites), which had been identified in various human cancers, esophageal adenocarcinoma, liver cancer, B-cell lymphoma, and stomach cancer[39–41], in iPSCs generated from MEFs by retrovirus-mediated gene transduction. Intriguingly, this signature was evident only in stage 2, which occurs just after the onset of cell lineage conversion during which bona fide de novo point mutations arise, but neither in pre-existing SNVs (stage 1) nor in the SNVs that arose during prolonged culture (stage 3) (Fig. 6a). These findings indicated that Signature 17 is a product of the genome-reprogramming process. It is also noteworthy that like the SNVs found to occur in stage 2, a clear Signature 17 pattern was also evident among the 50%VAF-SNVs which are the most abundant point mutations in most lines but cannot yet be experimentally demarcated as de novo or pre-existing (Fig. 6b). Significantly, Signature 17 was also observed in other mouse reprogrammed pluripotent stem cells, such as integration-free iPSCs, ntESCs, iPSCs induced by doxycycline (Dox)[14], and in iPSCs generated from other types of somatic cells (Supplementary Fig. 20a–c).

Importantly, Signature 17 was not found among pre-existing SNVs in the parent somatic cells or in the SNVs identified in ES cells (Fig. 6a), but was nevertheless observed in all reprogrammed pluripotent stem cells examined in this study, i.e. 23/23 iPSC lines and 11/11 ntESC lines (Supplementary Fig. 20a and b). More importantly, the cancer signature was also revealed in 3F (without c-Myc infection)-iPSCs (Supplementary Fig. 20a bottom). This strongly suggested that genome reprogramming and tumorigenesis pathways for some types of cancer have developmental processes in common[42]. In addition, an important finding of our current analysis was that Signature 17 was generated within a short period and could be re-created experimentally, thus offering a useful future system for elucidating the mechanisms underlying this cancer signature (Fig. 6a).

On the other hand, we could not detect Signature 17 in 33 human iPSC lines that had been generated by four independent groups including our laboratory, which contrasts with their mouse counterparts (Fig. 6c). This could not have been attributable to genetic differences as Signature 17 has been observed in human cancers[39]. Notably in this regard, the point mutations we detected in human iPSC genomes may indicate another cancer mutational signature, Signature 18, which is

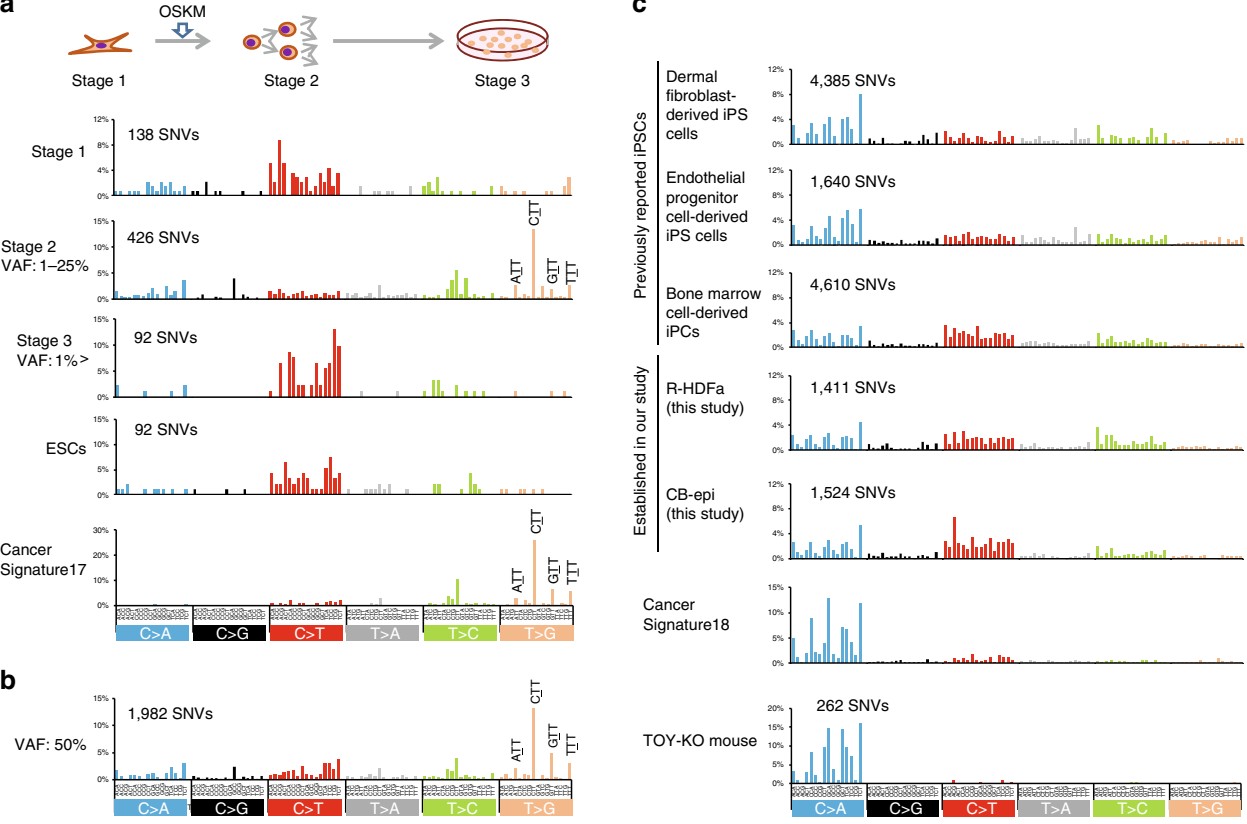

**Fig. 6 A cancer mutational signature exists in mouse reprogrammed pluripotent stem cells. a** For cancer mutational signature analysis, SNVs were divided into 96 classes in accordance with the base substitution pattern and flanking nucleotides[39]. The Y-axis shows the percentage of mutations attributed to a specific mutation type. (Upper) Timeline of iPSC generation: pre-stage (stage 1); mid-stage (stage 2); and post- stage (stage 3) reprogramming states. (Lower) Mutational signature at each step: (stage 1) pre-existing SNVs that have been reported as background mutations in parental cells[3]; (stage 2) point mutations that arose just after the onset of cell lineage conversion during iPSC generation, i.e. bona fide de novo variations (Fig. 5: 1–25% VAF); and (stage 3) point mutations that arose during prolonged culture (Fig. 5: 1% > VAF). The ESC results identifying SNVs in four mouse ES cell populations (B6ES2-2, B6ES2-7, B6ES2-9, and B6ES2-11)[4], and cancer mutational signature 17, are also shown[39]. **b** Mutational signature for the 50% VAF-SNVs in the R-4F-17, R-4F-28, and 2A-4F-136 iPSC lines, comprising 1982 SNVs in total (847, 564, 571 SNVs in R-4F-17, 2A-4F-136, and R-4F-28, respectively, which are indicated in the Venn diagrams shown in Fig. 5b). The 50%VAF indicates that the SNV exists in all cells within a colony because the mutation has basically arisen in either the paternal or maternal allele, but not both. Such 50%VAF SNVs are the most abundant point mutations in most lines derived from various reprogrammed pluripotent stem cells[3,4,14]. **c** Mutational signature analysis of human iPSC lines, including those previously reported. Details of human iPSCs are provided in Supplementary Table 2. Cancer Signature 18 has been previously described in various cancers, most notably in neuroblastoma[39]. TOY-KO mice, Mth1/Ogg1/Mutyh triple knockout mice[61].

similar to signature 8 and 5 (Supplementary Fig. 20d). We found several close similarities between human iPSCs and cancer cells in terms of 'C' to 'A' transversion base-substitution peaks (Fig. 6c).

**Influence of external stimuli on the mode of point mutations.** The issue of whether the nature of the point mutations (i.e. their frequency and base substitution profile) could be controlled and possibly reduced was of vital importance not only for elucidating the underlying molecular mechanisms of these mutation events, but ultimately for the future safe use of iPSCs in regenerative medicine. We examined this issue cytochemically and found an accumulation of 8-oxoG, which is a main causative factor for transversion base substitutions, at the early stages of iPSC generation using either Dox-inducible or human retroviral transduction systems (Fig. 7a, b and Supplementary Fig. 21). This finding was consistent also with the results of a previous report[43]. We next generated mouse iPSCs by introducing the repair genes for 8-oxoG, 8-oxoguanine DNA glycosylase 1 (Ogg1), mutY DNA glycosylase (Mutyh), apurinic/apyrimidinic endodeoxyribonuclease 2 (Apex2), and nudix hydrolase 1 (Nudt1) along with the Yamanaka factors, and then performed WGS on 5 of the

resulting clones (Fig. 8a and Supplementary Data 4). The results indicated that one of these clones had a transition-predominant base substitution profile (clones oG420, Fig. 8c). This was quite an intriguing finding because no other transition-predominant iPSCs have been described previously (9 control lines shown in Fig. 8c)[4,6].

When we conducted ROS induction at days 2–3 of iPSC generation using either IR or rotenone, which is an 8-oxoG producer[44], we obtained iPSCs harboring a larger number of point mutations than the control iPSC clones (Fig. 8b, c). Notably, we observed substantial numbers of SNVs (2081) in clone rot8115, which was an approximately three-fold higher number of point mutations than in the control iPSC clones. IR of the cells on day 2 also produced a dose-dependent increase in the number of point mutations. Of note, most of the increase in SNVs induced by rotenone and IR were 50% VAF, indicating that 50% of these SNVs were de novo and not pre-exiting (Supplementary Fig. 23). Intriguingly, strikingly similar phenomena were observed during human iPSC generation, though transversion base substitution is not as predominant in human iPSCs compared to their mouse counterparts (Supplementary Figs. 22 and 23).

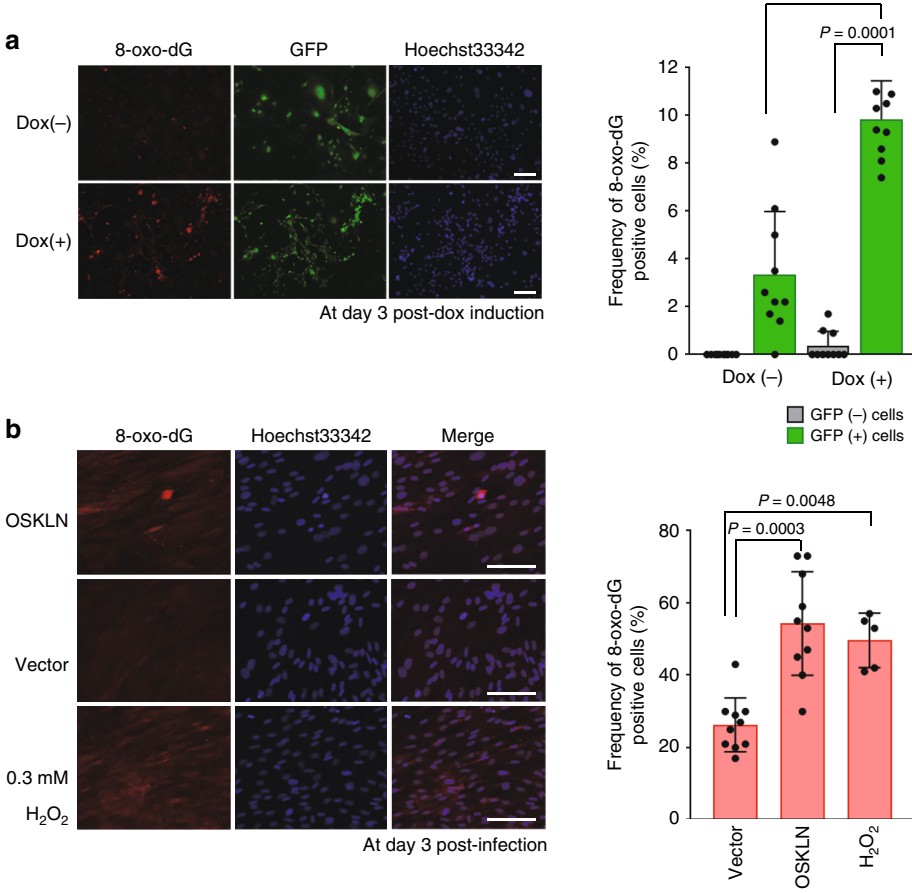

**Fig. 7 Accumulation of 8-oxoG in the initial stages of iPS generation. a** 8-oxo-dG staining at day 3 post-induction using Yamanaka factors (mouse Dox-inducible system). The MEFs used were derived from a chimeric mouse generated with a GFP-positive iPSC line containing the Dox-inducible transgene constructs encoding the four reprogramming factors (see Supplementary Fig. 9a). Scale bar, 100 μm. Error bars show the SD of the mean (n = 10 fields randomly selected of two independent dishes). For statistical analysis, the Mann–Whitney U-test was employed. **b** 8-oxo-dG staining at day 3 post-infection (human retroviral system). Human fibroblasts (BJ cells) were infected with OSKLN cloned into a retrovirus all-in vector. The method used for quantitative imaging is shown in Supplementary Fig. 21b. Scale bar, 100 μm. Error bars indicate the SD of the mean (n = 10 fields randomly selected of two independent dishes). For statistical analysis, the Mann–Whitney U-test was employed. Source data are provided as a Source Data file.

We next attempted to reduce the point mutation frequency in iPSCs by removing the ROS from the cells. We introduced the antioxidants, superoxide dismutase 1 (Sod1), superoxide dismutase 2 (Sod2) glutathione peroxidase 1 (Gpx1), and N-acetyl cysteine (NAC) along with the Yamanaka factor genes (Fig. 8a, c) and succeeded in generating an iPSC clone, NAC1-7, that had fewer and less potentially problematic SNVs, namely non-transversion-predominant SNVs (Fig. 8c). At this current stage, while only a small number of clones showed a mutation reduction under external stimuli, and even in the iPSCs in which reduction were observed it was only partial, a significant correlation was demonstrated between the numbers of SNVs reduced by external stimuli and base substitution profile change from a transversion- to a transition-predominance (Fig. 8d). Thus, the point mutations in iPSC genomes can be influenced by external stimuli, which may enable them to be appropriately controlled for future applications.

## Discussion

In addition to the phenomena discussed in the section "Introduction", another interesting observation led us to the hypothesis that formed the basis of this present study. When we previously conducted time-lapse analysis of iPSC generation, we observed very fast cell cycles (~7 h/cycle) just after the onset of the cell lineage conversion from a somatic cell to a stem cell[45,46]. Considering that the cell division time is usually prolonged when DNA damage occurs, this rapid cell cycle could not be readily explained. Our current findings clearly indicate that at days 2–3 post-infection during iPSC generation, a cell cycle checkpoint deficiency arises. If DNA repair was the only system deficiency to occur during this process, a radiation-sensitive phenotype leading to a reduced iPSC generation frequency would be expected[21,47]. In contrast, cell cycle checkpoint defects can cause radiation-resistance, in which an accumulation of genetic aberrations including point mutations may also occur[48,49]. Importantly, cell cycle checkpoint defects are typical of cancer cells, although they are constitutive and not transient in this case[38]. Hence, our observations led us to speculate that iPSCs may transiently experience tumorigenic-like events during genome reprogramming, a consideration which was strongly supported by our findings of cancer mutational signatures, '17' in mouse and '18' in human, in the reprogrammed pluripotent stem cell genomes. In support of this was our observation of a transient induction of cyclin D1 followed by Rb phosphorylation, events which are frequently observed in various cancers[50] although not transiently, at the early stages of iPSC generation. These phenomena were reprogramming-specific and not dependent on c-Myc infection. Signature 17 was observed in 3F-iPSCs (without c-Myc infection)

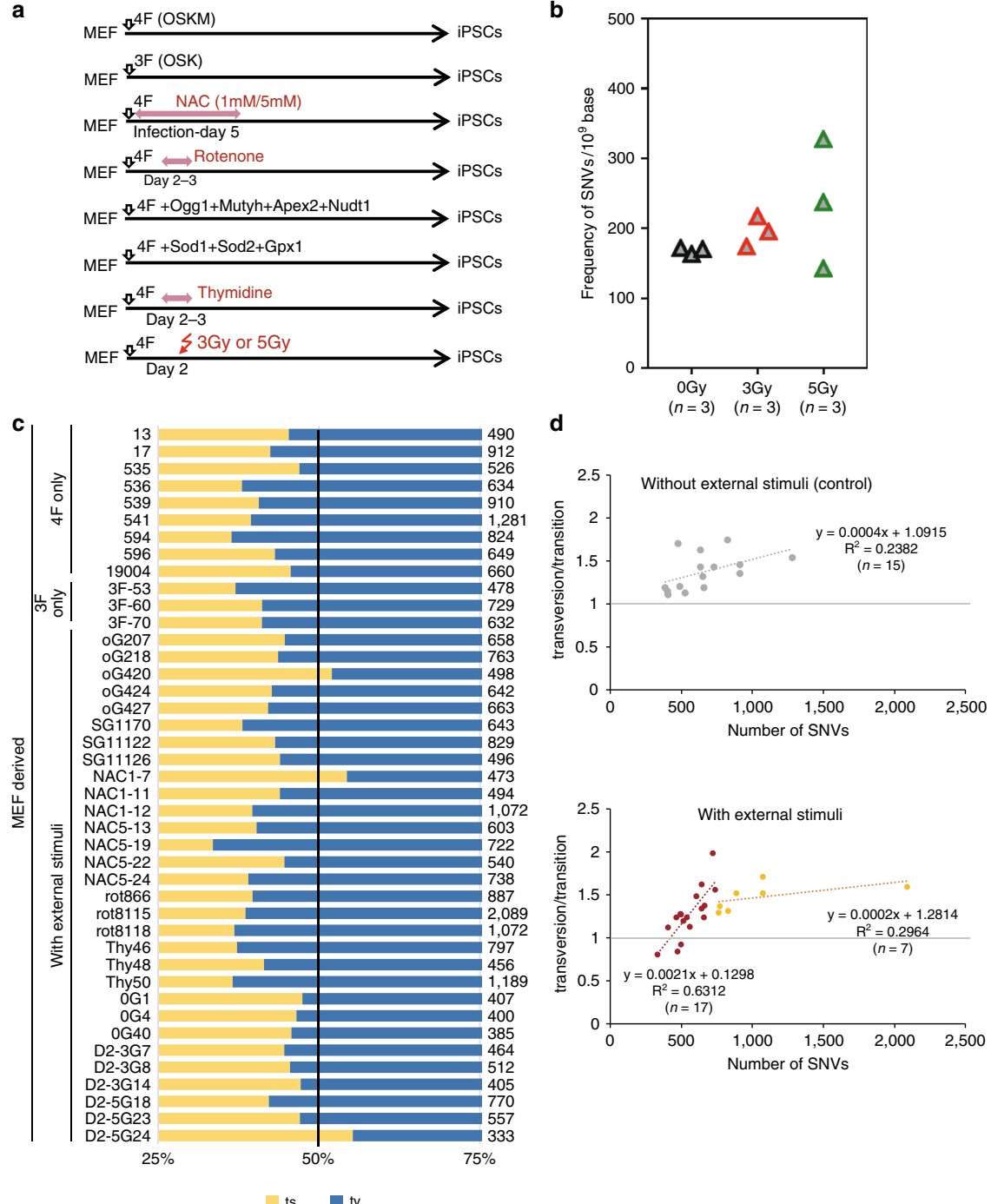

**Fig. 8 WGS analysis to assess the effect of external stimuli on the point mutation profile in mouse iPSCs. a** Experimental design for mouse iPSC generation. **b** Effects of IR on the formation of point mutations during iPSC generation (mouse). Cells were exposed to X-rays at 3 or 5 Gy on day 2 ($n = 3$ iPSC lines). **c** The tv/ts ratio and total number (right side of the panel) of identified SNVs for each clone are shown. To reduce the number of false-positives as much as possible, all iPSC generation experiments were conducted using single embryo-derived MEFs (MEF5). **d** Effect of external stimuli on the mode of point mutations in iPSC genomes. A high correlation ($R^2 = 0.6312$) was revealed between the number of SNVs reduced by external stimuli and the base substitution profile shift from a transversion- to a transition-predominance ($n = 7-17$ iPSC lines). Source data are provided as a Source Data file.

also and a boost of cyclin D1-Rb phosphorylation was not observed following c-Myc only infection (Fig. 3c). On the other hand, however, it must be noted that these results do not necessarily mean that these pluripotent stem cells are cancer-prone[2,3,5]. Further studies including longitudinal analyses will be needed to resolve this matter in the future.

It was notable that the elevated expression of cyclin D1, and the positive effect of cyclin D1 and Rb inactivation on iPSC generation[51], have been implicated previously in the rapid cell proliferation during the early stages of iPSC generation[52,53]. In our present analyses however, we observed a transient down-regulation of cell cycle checkpoint function in the early stages of iPSC generation by examining the radio-sensitivity of the cells. We used X-rays for this purpose, as it causes DNA damage. Transient cyclin D1-Rb phosphorylation activation was found to occur simultaneously with the transient checkpoint deficiency.

Taken together, our results most likely suggest that a drastic induction of cyclin D1 followed by the phosphorylation of Rb overshoots the basal activities governed by the p53 pathway and generates a p53 expression feedback to return the expression of cyclin D1 to normal levels and thereby causing a transient cell cycle checkpoint deficiency (Fig. 3c, Supplementary Fig. 13).

In contrast to cancer cells, the cell cycle checkpoint attenuation during iPSC generation is transient only. However, considering that the functions of the cell cycle checkpoint, including p53 activity, are vital for genomic and cellular integrity, this transient downregulation is still of considerable interest. The issues and questions that will need to be addressed in the future include the reason for the transient attenuation of p53-related functions during cell lineage conversion. Intriguingly, a similar transient reduction in p53 activity has been suggested to have a role in the regeneration phenomenon in amphibians and during early embryogenesis in the mouse[54,55]. While it is currently unknown why the cyclin D1-Rb phosphorylation pathway is boosted and cell cycle checkpoint activity is down-regulated transiently at the cell lineage conversion stage of genome reprogramming, in which a dynamic reorganization of the epigenome also occurs, the question of whether this transient checkpoint deficiency is essential for genome reprogramming must be addressed in future reports. These findings might also indicate that an epigenomic reorganization cannot be executed under normal cell cycle checkpoint conditions[54,56–60].

We have identified different mutation signatures, Signature 17 for mouse and 5, 8, and 18 for human (Fig. 6). This finding may be the result of a different intracellular environment and/or different mechanisms operating during the development of these cells. There are many possibilities that could be discussed in relation to the dissimilarities between mouse iPSCs and human iPSCs, such as differences in ROS regulation and DNA repair systems in the parent somatic cells and/or during iPSC generation. Most interestingly, a typical Signature 18 profile was identified by analyzing SNVs previously reported in the Mth1/Ogg1/Mutyh triple knockout (TOY-KO) mice[61,62] and Signature 17 was not detected at all. This result clearly demonstrated that 8-oxoG and/or 2-OH-dA are the causes of Signature 18 (Fig. 6c bottom). On the other hand, it suggested that a critical repair enzyme that causes Signature 17 was missing. This missing enzyme, including a proofreading activity, may have become non-functional during mouse iPSC generation but still be active in human iPSC generation. This could also be due to the well-known differences between the pluripotency states of mouse and human iPSCs, i.e. naïve and primed[63]. In any case, these cancer mutational signature differences, and the mechanisms involved, are quite intriguing.

In terms of the potential medical uses of iPSCs, point mutations of obviously of great concern. In our present study therefore, we attempted to reduce the number of point mutations in iPSC genomes, especially in human cells. It is significant that we eventually succeeded in generating genome-integration free iPSCs with only a small number of point mutations by employing erythroblasts as the parent somatic cells. This may well pave the way for the actual future use of iPSCs in medical applications, particularly transplantation therapy.

We got the idea to use erythroblasts from our investigations that external stimuli influence the mutation frequency and base substitution mode in iPSCs. These results are of great importance because whilst they revealed that mutations can arise during iPSC generation, they also showed that the mutation frequency was potentially controllable. It still appeared initially however that it would be extremely difficult to obtain almost mutation-free iPSCs even if the cells were exposed to external stimuli, such as repair gene transduction or antioxidant treatment. Indeed, only a small number of clones showed a mutation reduction under such stimuli, and even

in the iPSCs in which reduction were observed, although the base substitution profile changed from a transversion-predominance to a transition-predominance, the degree of reduction was only partial (Fig. 8d). We speculated that it would be extremely difficult using antioxidants to reproducibly and effectively reduce the basal ROS levels to an almost complete absence in mitochondria as they are constitutively generated in these organelles, i.e. a state of dynamic equilibrium, and that there would be a limit to how much the mutation frequency could be reduced. In addition, it is still unknown whether a transient reduction in ROS directly results in a reduced basal level of 8-oxoG in the genome. Therefore, to counter the constant generation of ROS via oxidative phosphorylation in mitochondria, we searched for somatic cells in which the energy metabolism is predominantly glycolytic, among the cells which can be prepared less invasively. In addition, in all of the erythroblast preparations examined, which had been used for iPSC generation, a markedly high expression level was observed for catalase (CAT: a key antioxidant enzyme responsible for the conversion of hydrogen peroxide to water and oxygen)[64], glutathione S-transferase mu 3 (GSTM3: an antioxidant enzyme that catalyzes the conjugation of reduced glutathione to various substrates)[65], apolipoprotein E (APOE: an inhibitor of lipid oxidation)[66], and apoliporotein M (ApoM: has potential antioxidant activity)[67] (Supplementary Fig. 1f). Thus, an approximately 5–10-fold, 10–20-fold, 100–130-fold, and 3–4-fold increase in CAT, GSTM3, APOE, and APOM, respectively, compared to the expression levels in human dermal fibroblasts (HDFs), was shown. Although the mechanisms underlying the genetic stability in erythroblasts will need to be elucidated in further studies, the unique expression profiles of antioxidant genes revealed in our present analyses indicate a constitutive low level of ROS in these cells. In additon, interestingly rather high expression was observed for OGG1. To be frank, the extent of the reduction in mutations was far beyond our expectations (Fig. 1). We believed that the erythroblast fraction expanded from CB would potentially be ideal somatic cells for this purpose. We consider that the molecular basis for this almost complete reduction in point mutations provides viable strategies for reducing the mutation frequency during iPSC generation. In addition, it may be also noteworthy that the number of SNVs can be reduced to some extent during their growth in culture, in which certain cells with a growth advantage become dominant within the iPSC clone[68].

Nevertheless, the OE of repair genes and/or treatment by ROS scavengers should be attempted further. In particular, an OE of catalase, for which a marked expression was observed in erythroblasts, and a longer incubation with NAC would likely be promising approaches. In contrast, the OE of repair components, such as glycosylase would not work effectively as it could cause severe genomic injury due to the unbalanced repair activities elicited by a higher expression of just one participant. The introduction of a set mismatch repair system genes instead of a single repair gene may be required in this circumstance. In addition, the use of metabolic inhibitors that shift the energy metabolism toward a glycolysis-predominant state appears also to be a promising approach.

Inhibitors of PDH, which is the gatekeeper between glycolysis and the TCA cycle, are good candidates in this regard, and use of pyruvate dehydrogenase complex deficiency (PDCD) cells would be a useful way to assess the reliability of this approach[69].

In our present study, we demonstrate a specific time-window during early iPSC reprogramming that is associated with the accumulation of point mutations in these cells and we describe the molecular mechanisms underlying this phenomenon. Deep sequencing is usually required to evaluate iPSCs for clinical use and sometimes reveals colony-merging events in an iPSC colony. This is a crucial issue in terms of the potential clinical application of these cells because these merging events can lead to more point

mutations. Indeed, colony-merging has been revealed in iPSCs by time-lapse[70] and was observed in our present analysis. We consider that most of the point mutations we observed in our current iPSC lines cannot be explained by colony-merging however, because current WGS technology, i.e. standard one run analysis with a ~40 read depth, basically only identifies the SNVs in dominant clones regardless of whether colony-merging has occurred. Nevertheless, the contribution of such events to the SNVs observed in iPSCs still remains elusive and more precise investigations should be conducted. In this regard, it is noteworthy that we also identified a substantial number of SNVs and INDELs in ntES cells, which is another reprogramming system that begins with single cell handling. This system could be useful for addressing this concern in a future study.

Any occurrence of colony-merging during iPSC generation must necessarily prohibit the subsequent medical use of these reprogrammed pluripotent stem cells. In this regard however, the generation frequency of our current human CB-erythroblast-derived integration-free iPSCs is sufficiently low at 0.0024%, in which we can generate iPSC at a frequency of single colony per culture dish, to avoid possible concerns regarding colony-merging.

## Methods

**Human cell cultures**. Mononuclear cells prepared from human cord blood (CB-MNCs) of four individuals were purchased from PromoCell (single donor, ultra-pure, cryopreserved, C-12901; Heidelberg, Germany). For erythroblast expansion, these cells were cultured for 8 days in serum-free medium (SFM) containing 50% IMDM with 50% Ham's F12, synthetic lipids, ITS supplement (Thermo Fisher Scientific, Waltham), 5 mg/ml BSA (Nacalai tesque, Kyoto, Japan), 50 μg/ml of ascorbic acid (Sigma-Aldrich), 2 mM glutamine (Thermo Fisher Scientific), IL-3 (10 ng/ml, Peprotech, Rocky Hill CT), SCF (50 ng/ml, Peprotech), EPO (2 U/ml, ESPO, Kirin Brewery Co. Ltd., Tokyo, Japan), IGF-1 (40 ng/ml, Peprotech), and dexamethasone (1 μM, Sigma)[71]. The expanded cells were validated by the detection of erythroblast marker expression (CD71 and CD235a) using flow cytometry. For expanding CD34+ cells, which were used for control, CB-MNCs were cultured for 8 days with a SFM containing 50% IMDM with 50% Ham's F12, synthetic lipids, ITS supplement (Thermo Fisher Scientific) and 5 mg/ml BSA (Nacalai tesque), 50 μg/ml of ascorbic acid (Sigma-Aldrich) and 2 mM glutamine (Thermo Fisher Scientific), IL-3 (10 ng/ml, Peprotech), SCF (100 ng/ml, Peprotech), Fit3-L (100 ng/ml, Peprotech), and TPO (20 ng/ml, Peprotech). Jurkat cells (RIKEN BRC, RCB3052) was cultured in RPMI supplemented with 10% FCS. Keratinocytes (newborn, male, FC-0007) were purchased from Lifeline Cell Technology (Frederick, MD) and cultured with HuMedia-KG2 medium (Kurabo). Two populations of human adult female dermal fibroblasts (HDFs) were purchased: FC-0024, Lifeline Cell Technology and C0135C, Thermo Fisher Scientific. These cells were cultured in 10% FCS–DMEM and in medium 106 (M-106-500, Thermo Fisher Scientific) supplemented with Low Serum Growth Supplement (S-003-10, Thermo Fisher Scientific), respectively. Informed consent has been obtained for use of the human primary cells in research.

**ROS assay**. The cells were collected and stained with 500 nM CellROX Deep Red (Thermo-Fisher Scientific) for the detection of intracellular ROS.

**Western blotting assay**. Cells were washed three times with 3 ml PBS, harvested by scraping into 1 ml PBS, and then centrifuged at $1000 \times g$ for 5 min at room temperature. The supernatant was then removed and the cell pellets were dissolved in sodium dodecyl sulfate (SDS) lysis buffer containing 62.5 mM Tris–HCl (pH 6.8), 2% SDS, 10% glycerol, complete protease inhibitor cocktail (Roche, Basel, Switzerland), and a Halt phosphatase inhibitor cocktail (Thermo Fisher Scientific). Homogenization of the cells was achieved by ultrasonication (Bioruptor; Cosmo-Bio, Tokyo, Japan) on ice until the cell debris was completely dissolved. Protein concentrations were measured using the bicinchonic acid (BCA) assay (Thermo Fisher Scientific). Aliquots of 10 or 20 μg of each extract was denatured in NuPAGE LDS sample buffer with heating for 10 min at 70 °C prior to loading onto the gels. These samples were then subjected to SDS–PAGE and transferred to a PVDF membrane using the iBlot Dry Blotting System (Thermo Fisher Scientific). Membranes were blocked with 5−7% skim milk-TBST for 1 h at room temperature or for overnight at 4 °C, and incubated with the primary antibody for 1 h at room temperature or for overnight at 4 °C followed by a secondary antibody for 1 h at room temperature (Supplementary Table 3). All antibodies used in the assay were appropriately diluted in Can Get Signal Immunoreaction Enhancer solution (Toyobo, Osaka, Japan). Immunoblot signals were detected via enhanced chemiluminescence (ECL select, GE Healthcare UK Ltd, Buckinghamshire, England, UK), and imaged using a Lumino image analyzer, LAS 4000 (Fujifilm, Tokyo, Japan). All uncropped blots are shown in the Source data.

**Microarray analysis of gene expression**. Total RNA was prepared using an RNeasy mini kit (Qiagen). Purified and labeled RNA was hybridized to an Agilent SurePrint G3 Human Gene Expression v3 8 × 60K Microarray (Design ID: 072363, Agilent, Inc., Santa Clara, CA, USA) in accordance with the manufacturer's instructions. Microarray results were extracted using Agilent Feature Extraction software v11.0 (Agilent Technologies).

**Generation of integration-free iPSCs from erythroblasts**. iPSCs were generated using a polycistronic expression episomal vector, in which Oct3/4, Sox2, Klf4, c-Myc, and Lin28 were cloned, as previously reported (pEB-C5, Addgene)[17,71].

To generate iPSCs from the expanded erythroblasts, pEB-C5 was transfected into $1 \times 10^7$ of these cells (at 10 μg per $2 \times 10^6$ erythroblasts) using the Human CD34 nucleofector kit (VPA-1003, Lonza, Basel, Switzerland) under the program T-016 (Amaxa, Lonza). After transfection, the cells were cultured with the above-mentioned SFM for further 2 days and then transferred onto a human vitronectin-coated 12-well plate (VTN-N, A14700, Thermo Fisher Scientific) (for feeder-free culture) or a MEF-coated 12-well plate ($1 \times 10^5$ cells/well) followed by centrifugation at $200 \times g$ for 30 min, and cultured with MEF medium (10% FBS, DMEM) (culture condition [8] shown in Supplementary Fig. 6). At the next day (day 3) MEF medium was replaced with ESC medium (Essential 8 medium, from Thermofisher Scientific) and exchanged it every other day, and on day 9 ESC medium was replaced with MEF-conditioned medium (CM; MEF cultured E8 medium) supplemented with 10 ng/ml basic fibroblast growth factor (bFGF from Repro Cell). Of note, sodium butyrate (0.25 mM) (NaB; TR-1008-G, Merck Millipore, MA, USA) was added from day 2 to day 21. We picked colonies at day 14–21, and maintained in E8 medium with recombinant human vitronectin (VTN-N, A14700, Thermofisher Scientific)-coated plates.

**Assessment of genome integration-free status**. Genome DNA was prepared using a DNeasy mini kit (Qiagen) for subsequent PCR analysis using 20 ng aliquots (corresponding to about 3000 cells) in each reaction. The amplification conditions were 1 cycle at 94 °C for 2 min followed by 35 cycles at 94 °C for 30 s, 55 °C for 30 s, and 72 °C for 1 min with 1 final cycle for 7 min at 72 °C. For the β-globin amplification control, the protocol was 1 cycle at 94 °C for 30 s followed by 35 cycles at 94 °C for 30 s, 55 °C for 2 min, and 72 °C for 1 min. The primer sequences were as follows: 5′-TTC CAC GAG GGT AGT GAA CC-3′ and 5′-TCG GGG GTG TTA GAG ACA AC-3′ for oriP, 5′-ATC GTC AAA GCT GCA CAC AG-3′ and 5′-CCC AGG AGT CCC AGT AGT CA-3′ for EBNA1, 5′-TCG CTC CAG TCA ATG ACC GC-3′ and 5′-CGT TAT GTT TAT CGG CAC TTT GC-3′ for hygromycin and 5′-GGT TGG CCA ATC TAC TCC CAG G-3′, and 5′-CAA CTT CAT CCA CGT TCA CC-3′ for β-globin. Furthermore, verification was also performed using WGS data for iPSCs. We selected the reads that were not aligned (mapped) to the human reference genome sequence and examined the sequence of the vector used for gene transduction (pEB-C5, Addgene).

**Immunocytochemical staining of pluripotent markers**. Cells were fixed with 4% phosphate-buffered paraformaldehyde for 15 min and washed three times with PBS. For the detection of intra-cellular antigens, the fixed cells were permeabilized using 0.1% Triton X-100 for 15 min and washed three times with PBS. After blocking with PBS containing 1% BSA for 60 min, the samples were incubated in primary antibodies at 4 °C overnight (Supplementary Table 3). Nuclei were counterstained with Hoechst33342 (1:2000, 346-07951, Dojindo, Kumamoto, Japan). The stained samples were embedded in Fluoromount/Plus (Diagnostic Biosystems Pleasanton) and observed under a fluorescence microscope (BZ-9000, Keyence).

**In vitro assay for differentiation abilities**. To generate embryoid bodies (EB), iPSCs were cultured with differentiation medium (50% F-12, 50% DMEM, 20% FCS, 1% ITS solution) in an ultra-low attachment six-well plate (ultra-low attachment surface plate, 3471; Corning, New York, NY, USA) for 8 days. The EBs were then transferred to gelatin-coated chamber slides (μ-slide eight-well, ibi-treat, ib80826; Ibidi, Martinsried, Germany) and cultured for a further 2 days to enable cellular attachment and differentiation. To investigate their differentiated states, the cells were washed with PBS and fixed with 4% phosphate-buffered paraformaldehyde for 30 min, and then washed three times with PBS. Thereafter, the fixed cells were incubated with PBS containing 3% BSA and 0.1% TritonX-100 for 30 min. For blocking and permeabilization, the plates were washed three times with 0.1% Triton X-100/PBS, and then incubated with each primary antibody at 4 °C overnight (Supplementary Table 3). After further washing three times with 0.1% TritonX-100/PBS, the cells were reacted with secondary antibodies at room temperature for 60 min.

**WGS and identification of SNVs**. Genomic DNAs were prepared using DNeasy (Qiagen) for use in WGS analysis. The procedures employed in this study, including the sequencing reactions, mapping, and SNV calling systems, are summarized in Supplementary Data 1 and 4. Other than in a small portion of the sublines, WGS libraries were prepared using the TruSeq PCR-free DNA-prep kit and sequenced with HiseqX with 150 bp paired-end analysis.

                                                                                     

The obtained reads were mapped to the C57BL/6 mouse reference genome (NCBI37/mm9) or human reference genome (GRCh37/hg19). We allowed up to 2% (length fraction, 1; similarity fraction, 0.98 for C57BL/6 mouse iPSCs) or 3% (length fraction, 1; similarity fraction, 0.97 for Nanog-GFP mouse iPSCs and human iPSCs) mismatches in the mapping and only uniquely mapped reads were used in the subsequent analysis. SNV candidates were identified using the CLC Genomics Workbench with the following parameters: minimum quality of central base, 30; minimum average quality of surrounding bases, 15; window length, 11; maximum gap and mismatch count, 2; and depth ≥20. The following SNVs were removed from our panel of candidates: shared with the parental somatic cells (number of variant alleles in parental cells ≥ 2); known variants (dbSNP128 or dbSNP130); shared among sister clones; or had flanking homopolymeric or simple repeat bases. SNVs with a VAF of 35–65% were counted for autosomes, and those with a VAF ≥ 75% for the male sex chromosome. Our current study methodology, including the bioinformatic analyses, included verification by Sanger sequencing as described previously[4]. To compare our current iPSCs against other previously reported human iPSCs, previously reported WGS data were downloaded from public databases and reanalyzed using our own bioinformatics approaches.

**Identification of INDELs in human iPSC genomes**. INDEL analysis was also conducted with the WGS reads used for SNV analyses. The mapping and calling conditions (CLC genomics workbench) were as follows. Mapping: reference genome, GRCh37/hg19; length fraction, 0.5; similarity fraction, 0.97; non-specific match handling, ignore. Variant detection: coverage, ≥20; minimum count, 10; ignore broken pairs, true.

The following candidates were removed from our panel: shared with the parental somatic cells (number of variant alleles in parental cells≥2); known variants (dbSNP150-common); shared among sister clones; or had flanking homopolymeric or simple repeat bases. INDELs with a VAF of 30–70% (for male sex chromosomes, VAF ≥ 70%) were counted. We also verified our candidates. Twenty-three INDEL candidates, which were whole candidates detected in the four human iPSC clones CB-epi-1-1, epi-1-5, epi-1-8, epi-1-11, were examined. Twenty out of the 23 candidates were successfully analyzed by Sanger sequencing, and 19 (95%) were finally confirmed to be correct (Supplementary Fig. 3)

**SNP array analysis**. Genomic DNAs were prepared using DNeasy (Qiagen). An Infinium Omni2.5-8 v1.4 BeadChip (Illumina) was used for detecting CNVs.

**Generation of mouse iPSCs**. Nanog-GFP tg [STOCK Tg(Nanog-GFP, puro) 1Yam] (RIKEN BRC) mice[20] were mated with C57BL/6J (Japan SLC, Hamamatsu, Japan) mice and MEFs were prepared and pooled from embryonic day 13.5 embryos. For retroviral infection, pMX vectors containing Oct3/4, Sox2, c-Myc, and Klf4 (Addgene, Cambridge, MA, USA) were transfected into PlatE cells using FuGENE 6 (Promega, Madison, WI, USA)[19]. The medium was collected at 48 h post-transfection, filtered, supplemented with polybrene (4 μg/ml), and added to MEFs cultured in six-well plates at $6.75 \times 10^4$ cells/well. The cultures were incubated for 16 h for infection to take place and the medium was then replaced with DMEM supplemented with 15% FCS, leukemia inhibiting factor (LIF), penicillin/strepto-mycin, L-glutamine, nonessential amino acids, and 2-mercaptoethanol (2-ME). The 15% FCS supplement was subsequently replaced with 15% knockout serum replacement (KSR) (Invitrogen, Carlsbad, CA, USA) at day 4 or day 7 post-infection (Supplementary Fig. 6 [1] or [2])[45,72]. The mouse iPSC line, 1B, was generated from ROSA26 knock-in rtTA-IRES-GFP MEFs using *piggyBac* vectors containing the Dox-inducible reprogramming factors, Oct4, Sox2, Klf4, and c-Myc[34,35]. The cells were cultured on mitomycin C-treated MEFs in DMEM containing 15% FCS, LIF, penicillin/streptomycin, L-glutamine, nonessential amino acids, sodium pyruvate, and 2-ME. To establish secondary iPSCs, we prepared MEFs from 1B cells. Briefly, these cells were injected into BDF1 embryos (8-cell embryos–blastocysts) to prepare chimeras. MEFs were subsequently prepared at day 13.5. The donor cell contribution for each MEF was assessed by GFP expression using flow cytometry. The obtained MEFs were cultured ($1.5 \times 10^4$ cells/well on feeder cells in a six-well plate) in DMEM with 10% FCS one day before dox treatment and then in KSR-ES medium supplemented with doxycycline (days 1–8: 1500 ng/ml and days 9–12: 5 ng/ml)[35] (Supplementary Fig. 6 [3] or [4]). The animals were housed under a controlled lighting condition (daily light 07:00–19:00 h) and used at 8–12 weeks (female) and 8–40 weeks (male). All mouse experiments were performed in accordance with the relevant guidelines and were approved by the Animal Care and Use Committee of the National Institute of Quantum and Radiological Science and Technology, and University of Yamanashi.

**X-ray irradiation of cells with Wortmannin and EdU treatments**. The cells were irradiated with 3 Gy (dose rate 1.1 Gy/min) using the Pantak HF-320S X-ray apparatus (200 kVp, Shimadzu Co., Tokyo, Japan). Wortmannin (R&D systems, Minneapolis, MN, USA) was added at a final concentration of 10 μM at 15 min prior to X-ray irradiation and removed 24 h later. EdU (5-ethynyl-2′-deoxyuridine, Thermo Fisher Scientific, Waltham, MA) was added at a final concentration of 10 μM at 12 h after X-ray irradiation, and then removed 12 h later. GFP-positive colonies were counted at day 14 under a fluorescence microscope (BZ-9000 and BZ-X, Keyence, Tokyo, Japan). Colonies were picked on days 13–15. In the Dox-

inducible system and human iPSC generation system, alkaline phosphatase-positive colonies were counted.

**Human iPSC generation system using Sendai virus vectors**. BJ human foreskin fibroblast cells (ATCC CRL-2522) were seeded into 12-well plates ($5.15 \times 10^4$ cells/well) on Laminin-511E8 (iMatrix, Nippi, Tokyo, Japan) in fibroblast culture medium. Two days later, the fibroblasts were incubated with the CytoTune-iPS 2.0 Sendai Reprogramming Kit (ID Pharma, Tokyo, Japan) containing four Sendai virus vectors, OCT3/4, SOX2, KLF4, and C-MYC (MOI = 5)[73], for 24 h. The fibroblast medium was replaced with StemFit AK02N (Ajinomoto, Tokyo, Japan) on day 3 which was freshly replaced every other day (Supplementary Fig. 6 [5]).

**Human iPSC generation system using retroviral gene transduction**. For the preparation of the retroviral gene transduction solution, pDON-5 OKSLN, pDON-5, and pMXs-DsRed2 constructs were transfected together with pGP and pE-ampho vector genes into G3T-hi cells using TransIT-293 transfection reagent (Takara Bio Inc., Shiga, Japan). After 48 h, the medium was collected and filtered (0.45 μm pore size). To establish the iPSC lines, human dermal fibroblast cells (FC-0024, Lifeline Cell Technology, Frederick, MD) were infected with the viruses with RetroNectin in accordance with the manufacturer's protocol (Takara Bio Inc.) (Supplementary Fig. 6 [6]). For the X-ray irradiation assay, the virus was concentrated 10-fold using a Retro-X Concentrator (Takara Bio Inc.). BJ cells ($3.38 \times 10^4$ cells/well, 12-well plate) were then incubated with 0.25 ml of the concentrated virus solution and 0.75 ml of fresh fibroblast medium containing polybrene for 24 h followed by growth in fibroblast medium supplemented with 10% FCS-DMEM. Thereafter, the cells were replated onto feeder-coated ⌀10 cm dishes at day 7 and incubated for 24 h. On the following day (day 8) the growth medium was replaced with Primate ES medium (ReproCELL, Kanagawa, Japan) supplemented with 5 ng/ml recombinant human basic fibroblast growth factor (bFGF, ReproCELL) (Supplementary Fig. 6 [7]).

**Quantitative PCR analysis**. Total RNA was prepared using an RNeasy mini kit (Qiagen, Venlo, The Netherlands). Reverse transcription was performed using SuperScript III (Thermo Fisher Scientific) with an oligo-dT primer. Real-time PCR analysis was performed using SYBR Premix Ex Taq II (Takara Bio Inc.) and a StepOnePlus real-time PCR system (Thermo Fisher Scientific). The reaction conditions were as follows: 1 cycle at 95 °C for 5 s, 50 cycles at 95 °C for 5 s and 60 °C for 30 s. The data were normalized to the expression level of the Gapdh gene. The primer sequences were as follows: 5′-TCT CAG GGC CGA AAA CGG AG-3′ and 5′-ACA CAG AGT GAG GGC TAA GG-3′ for *p21* and 5′- GGA TGT GAA GGA TGG GAA GT-3′ and 5′- CCC TCT ATG GGC TCG AAT TT-3′ for *Rps18*.

**Cell cycle analysis**. A Click-iT Plus EdU Flow Cytometry Assay kit (Alexa Flour 647 picolyl azide, Thermo Fisher Scientific) was used to conduct the cell cycle analysis with a Vybrant DyeCycle Violet Stain (Thermo Fisher Scientific). Cells were incubated for 2 h with Edu at 6 h or at 24 h post-irradiation, followed by the Click reaction according to the method recommended by the supplier. The cells were then analyzed using a CytoFlexS device with CytExpert software (Beckman Coulter, Fullerton).

**Annexin V assay**. The 4F-infected cells exposed to X-ray irradiation on days 3 or 6, followed by 24 h of culture, were incubated with Accutase for 7 min (Innovative Cell Technologies Inc., San Diego, CA, USA), collected and then stained with a Dead Cell Apoptosis Kit with annexin V Alexa Fluor 488 & PI (Thermo Fisher Scientific) in accordance with the manufacturer's instructions for flow cytometry analysis (CytoFLEX S with Kaluza Analysis Software; Beckman Coulter).

**Establishment of sublines from single cells**. Single cell-derived sublines were generated as previously reported[4]. Briefly, to generate these sublines, single cells were selected from trypsinized iPSCs and ntESCs using a microcapillary pipette under a microscope and transferred into 96-well plates containing feeder cells. The culture conditions were 37 °C in a humidified environment of 5% $CO_2$ and sublines were obtained after 14–16 days. Genomic DNA was prepared from these sublines using a DNeasy mini kit (Qiagen) for subsequent WGS and amplicon sequencing. WGS libraries were prepared using a TruSeq DNA-prep kit or TruSeq PCR-free DNA-prep kit and sequenced with Hiseq 2500 with 101 bp paired-end or HiseqX with 151 bp paired-end analysis (Supplementary Data 5).

**Ultra-deep VAF analysis**. The genomic region containing the target SNV site was amplified by PCR in a reaction containing 15 ng of genomic DNA prepared from iPSCs (2500 cells). PCR was performed using Titanium-taq DNA polymerase (Clontech, Palo Alto, CA) under the following conditions: 95 °C for 1 min followed by 32 cycles at 95 °C for 20 s, 68 °C for 30 s, and 72 °C for 30 s. The PCR products were mixed and purified using a MinElute PCR Purification Kit (Qiagen), and then subjected to comprehensive sequencing using a HiSeq 2500 sequencer (101 bp, paired-end, Illumina). The VAF of each SNV was estimated by BWA mapping of the obtained reads. Primer sequences are listed in Supplementary Data 5.

**Immunostaining of 8-hydroxy-2′-deoxyguanosine**. Cells were cultured with ibidi glass bottom dishes (Martinsried, Germany), and then fixed in 4% formaldehyde and stained with anti-8-oxo-dG antibody [N45-1] (1:200, Abcam, Cambridge, UK, ab48508) and anti-GFP polyclonal antibody-Alexa Fluor 488 (1:200, Thermo Fisher Scientific, A-21311). Alexa Fluor 647 goat anti-mouse IgG was used as the secondary antibody. Images were analyzed using ImageJ (NIH image).

**Assessment of oxidative stress-related genes**. 8-oxoguanine repair genes (*Ogg1*, *Apex2*, *Mutyh,* and *Nudt1*) and antioxidant enzyme genes (*Sod1*, *Sod2*, and *Gpx1*) were isolated from mouse ESC line cDNAs by PCR, and cloned into the pMXs retroviral vector. Their sequences were then verified using the Sanger method. iPS clones were established using this retroviral system with either the four 8-oxoguanine repair genes or the three antioxidant enzyme genes in addition to the Yamanaka 4 factors. We verified the integration of all additionally infected genes in the obtained iPSC clones. Rotenone (final concentration 0.2 μM; Sigma-Aldrich, R8875) was added into culture medium from day 2 to day 3 post-infection with the Yamanaka 4 factors. NAC (final concentration 1 or 5 mM; Sigma-Aldrich, A9165) was added to the culture medium from just after infection to day 5, and this supplemented medium was changed every 24 h. Thymidine (final concentration 30 μM; Sigma-Aldrich, T1895) was added to the culture medium from days 2 to 3. All iPSCs were established from a single embryo (C57BL/6J)-derived MEF.

**Quantification of 8-hydroxydeoxy guanosine with ELISA**. Genomic DNAs were isolated from the cells with a DNA Extractor TIS kit (Wako Pure Chemical Industries, Osaka, Japan) and digested to mononucleotides with nuclease P1 using an 8-oxo-dG Assay Preparation Reagent Set (Wako Pure Chemical Industries). The 8-oxo-dG levels were analyzed with a high sensitivity ELISA kit (NIKKEN SEIL, Tokyo, Japan)[74].

**Reporting summary**. Further information on research design is available in the Nature Research Reporting Summary linked to this article.

## Data availability

The authors declare that all data supporting the findings of this study are available within the article and its Supplementary Information files or from the corresponding author upon reasonable request. The Raw Illumina sequencing reads generated during the current study have been deposited in the DDBJ Sequence Read Archive (DRA) under accession codes: DRA002912, DRA005034, DRA005296, DRA006232, DRA006233, DRA006234, DRA006457, DRA006458, DRA006622, DRA007325, DRA007336, DRA008453, DRA008459, and DRA009220. The previously generated Raw Illumina sequencing reads analyzed during the current study have also been deposited in the DRA under accession codes DRA000524, DRA002956, and DRA003544. SNP array data have been deposited in the DDBJ under accession code: E-GEAD-311. Microarray data have been deposited in the Gene Expression Omnibus (GEO) database under accession code: GSE131648. The source data for Figs. 2a, 3a, 4a–c, e, 3c, 7a, 7b, 8d and Supplementary Figs. 1c, 8, 9b, 11a–c, 14, 15, 23 are provided in the Source data file. All other data supporting the findings of this study are available from the corresponding author on reasonable request.

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

## Acknowledgements

We are grateful to S. Yamanaka (University of Kyoto) for providing the OSKM genes expressed in plasmid and retroviral vectors through Addgene and to T. Kitamura (University of Tokyo) for providing the PlatE cells. We also thank A. Fujimori for support with the X-ray irradiation experiments and Y. Murakawa, K. Matsumoto, T. Yasuda, M. Suzuki, and S. Hasegawa for helpful discussions. We further thank H. Yoshida, A. Ishibashi, and Y. Shindo for technical assistance. This work was partially funded by the Japan Society for the Promotion of Science (Grant nos. KAKENHI 25290068, 17H03615, 17K20053, and 16H06153) and by the Takeda Science Foundation, Uehara Memorial Foundation, and the Mitsubishi Foundation.

## Author contributions

R.A. designed the experiments, performed the computational analysis, and wrote the manuscript. Y.H. and C.O. performed most of the experimental work, including iPSC generation, the X-ray sensitivity assay, Wortmannin assay, and the cell cycle analysis. K.I. and M.F. performed the western blot assays. M.S. performed the VAF and quantitative RT-PCR analysis. T.S. performed the bioinformatics analysis. M.N. conducted the mouse experimental work. A.N. generated the Dox-inducible iPS cells. S.W. and T.W generated the ntESCs and the MEFs from Dox-inducible iPS cells. M.A. designed the experiments, analyzed the data, and wrote the manuscript.

## Competing interests

The authors declare no competing interests.
