## [Peer Review File · Nature Communications]

Editorial Note: This manuscript has been previously reviewed at another journal that is not operating a transparent peer review scheme.

Reviewers' Comments:

Reviewer #1:

Remarks to the Author:

I appreciate the efforts taken by the authors to improve the manuscript, which offers an interesting observation that I agree the field should be alerted to, and also to respond to reviewer comments. I provide my feedback to each of these in turn.

Reviewer 1: Could VAF-SNVs be due to colony merging rather than clonal acquisition during reprogramming in culture?

The authors cite technical reasons for why deep WGS might be necessary to more firmly establish if the VAF-SNV signature might be best explained by colony merging. While a fair point, it does not eliminate the concern that erythroblasts and fibroblasts might have an intrinsically different reprogramming efficiencies or likelihood to form mosaic colonies due to known cell migration on the culture dish during the reprogramming process. This potential technical explanation of their results should ideally be directly addressed, or at least discussed in detail in the Discussion.

Reviewer 1: Can pharmacological induction of glycolysis or reduction of ROS phenocopy lower mutation rates on non-erythroblast cell types?

I agree that rotenone is probably a poor choice due to its ROS induction, but it is unfortunate that there do not appear to be any pharmacological means to extend their findings to non-erythroblast cell types. Since pursuing this line of investigation could be of practical (and indeed commercial) interest to the field, it might be fruitful to add a discussion of potential means of extending their findings.

Reviewer 2: what is the mechanism by which ROS And 8oxoG contribute to mutations during reprogramming?

The authors argue that performing such experiments are technically challenging, but perform additional analysis showing increase expression of anti-oxidant genes. Overexpression of some of these genes would add value and credence to the study.

Reviewer 3: (no further comments)

Reviewer #1 (Remarks to the Author):

I appreciate the efforts taken by the authors to improve the manuscript, which offers an interesting observation that I agree the field should be alerted to, and also to respond to reviewer comments. I provide my feedback to each of these in turn.

We thank the reviewer for providing important insights and comments on our study.

Reviewer 1: Could VAF-SNVs be due to colony merging rather than clonal acquisition during reprogramming in culture?

The authors cite technical reasons for why deep WGS might be necessary to more firmly establish if the VAF-SNV signature might be best explained by colony merging. While a fair point, it does not eliminate the concern that erythroblasts and fibroblasts might have an intrinsically different reprogramming efficiencies or likelihood to form mosaic colonies due to known cell migration on the culture dish during the reprogramming process. This potential technical explanation of their results should ideally be directly addressed, or at least discussed in detail in the Discussion.

We fully agree that colony-merging occurs during iPSC generation and could be a serious obstacle to their medical use. However, deeper sequencing is required to assess this as the reviewer has mentioned. In addition, we also consider that erythroblasts and fibroblasts have an intrinsically different reprogramming efficiency indeed as the reviewer suggested.

We provide commentary on this in our revised Discussion as follows:

Page 24, line 14

“In our present study, we demonstrate a specific time-window during early iPSC reprogramming that is associated with the accumulation of point mutations in these cells and we describe the molecular mechanisms underlying this phenomenon. Deep sequencing is usually required to evaluate iPSCs for clinical use and sometimes reveals colony-merging events in an iPSC colony. This is a crucial issue in terms of the potential clinical application of these cells because these merging events can lead to more point mutations. Indeed, colony-merging has been revealed in iPSCs by time-lapse⁷⁰ and was observed in our present analysis. We consider that most of the point mutations we observed in our current iPSC lines cannot be explained by

colony-merging however, because current WGS technology, i.e. standard one run analysis with a ~40 read depth, basically only identifies the SNVs in dominant clones regardless of whether colony-merging has occurred. Nevertheless, the contribution of such events to the SNVs observed in iPSCs still remains elusive and more precise investigations should be conducted. In this regard, it is noteworthy that we also identified a substantial number of SNVs and INDELs in ntES cells, which is another reprogramming system that begins with single cell handling. This system could be useful for addressing this concern in a future study.

Any occurrence of colony-merging during iPSC generation must necessarily prohibit the subsequent medical use of these reprogrammed pluripotent stem cells. In this regard however, the generation frequency of our current human CB-erythroblast derived integration-free iPSCs is sufficiently low at 0.0024%, in which we can generate iPSC at a frequency of single colony per culture dish, to avoid possible concerns regarding colony-merging.”

Reviewer 1: Can pharmacological induction of glycolysis or reduction of ROS phenocopy lower mutation rates on non-erythroblast cell types?

I agree that rotenone is probably a poor choice due to its ROS induction, but it is unfortunate that there do not appear to be any pharmacological means to extend their findings to non-erythroblast cell types. Since pursuing this line of investigation could be of practical (and indeed commercial) interest to the field, it might be fruitful to add a discussion of potential means of extending their findings.

We agree that an approach using metabolic inhibitors warrants investigation, and that the creation of iPSCs with a low genetic burden from cells other than erythroblasts will be needed. We address these points in the revised Discussion as follows:

Page 24, line 9

“In addition, the use of metabolic inhibitors that shift the energy metabolism toward a glycolysis-predominant state appears also to be a promising approach. Inhibitors of PDH, which is the gatekeeper between glycolysis and the TCA cycle, are good candidates in this regard, and use of pyruvate dehydrogenase complex deficiency (PDCD)⁶⁹ cells would be a useful way to assess the reliability of this approach.”

Reviewer 2: what is the mechanism by which ROS And 8oxoG contribute to mutations during reprogramming?

The authors argue that performing such experiments are technically challenging, but perform additional analysis showing increase expression of anti-oxidant genes. Overexpression of some of these genes would add value and credence to the study.

We thank the reviewer for these helpful comments. It will be important to evaluate this in future studies but we hope that the reviewer will understand that the scale of these experiments are well beyond the scope of our present analysis.

We provide some commentary of the important issue in the revised Discussion as follows:

Page 24, line 3

“Nevertheless, the overexpression (OE) of repair genes and/or treatment by ROS scavengers should be attempted further. In particular, an OE of catalase, for which a marked expression was observed in erythroblasts, and a longer incubation with NAC would likely be promising approaches. In contrast, the OE of repair components such as glycosylase would not work effectively as it could cause severe genomic injury due to the unbalanced repair activities elicited by a higher expression of just one participant. The introduction of a set mismatch repair system genes instead of a single repair gene may be required in this circumstance.”

Reviewer 3: (no further comments)